**Spatial accessibility of emergency medical services under inclement**
**weather: A case study in Beijing, China**
Yuting Zhang [a], Kai Liu [a,b]*, Xiaoyong Ni [a,b], Ming Wang [a,b], Jianchun Zheng[c],
Mengting Liu[c], Dapeng Yu[d]
[a] *School of National Safety and Emergency Management, Beijing Normal University,*
*Beijing 100875, China*
[b] *Academy of Disaster Reduction and Emergency Management, Ministry of Emergency*
*Management & Ministry of Education, Beijing Normal University, Beijing 100875,*
*China*
[c] *Beijing Research Center of Urban System Engineering, Beijing 100035, China*
[d] *Geography and Environment, Loughborough University, Loughborough, UK*
* Corresponding author. E-mail address: liukai@bnu.edu.cn (Kai Liu); Full postal
address: School of National Security and Emergency Management, Beijing Normal
University, #19 Xinjiekou Wai Ave., Beijing 100875, China.

## Abstract

The accessibility of emergency medical services (EMSs) is not only determined by
the distribution of emergency medical facilities but is also influenced by weather
conditions. Inclement weather could affect the efficiency of the city's traffic network
and further affect the response time of EMSs, which could therefore be an essential
impact factor on the safety of human lives. This study proposes an EMS-accessibility
quantification method based on selected indicators, explores the influence of inclement
weather on EMSs accessibility, and identifies the hotspots that have difficulty accessing

timely EMSs. A case study was implemented in Beijing, which is a typical megacity in China, based on the ground-truth traffic data of the whole city in 2019. The results show that inclement weather has a general negative impact on EMSs accessibility. Under inclement weather scenario, the area in the city that could get EMSs within 15 minutes would decrease by 13% compared to normal scenario (the average state of weekdays without precipitation), while in some suburban townships, the population that could get 15-min EMSs would decrease by 40%. We found that snowfall has a greater impact on the accessibility of EMSs than rainfall. Although on the whole, the urban area would have more traffic speed reduction, towns in suburban with lower baseline EMSs accessibility are more vulnerable to inclement weather. Under the worst scenario in 2019, 12.6% of population (about 3.5 million) could not get EMSs within 15 minutes, compared to 7.5% with the normal condition. This study could provide a scientific reference for city planning departments to optimize traffic under inclement weather and the site selection of emergency medical facilities.

**Keywords**

Emergency medical services (EMSs), spatial accessibility, inclement weather, service area coverage

## 1 Introduction

Emergency medical services (EMSs) are a pivotal part of the public health system, and the response time of EMSs is a vital factor in decreasing morbidity and improving survival (Blackwell and Kaufman, 2002). In China, the EMS system is mainly composed of prehospital emergency services and in-hospital emergency services. Prehospital emergency service refers to on-site emergency treatment, guardianship in transit, and handover with in-hospital emergency institutions. The efficiency of emergency services is highly vulnerable to inclement weather conditions such as rain, snow, frog, etc. The reason why inclement weather conditions would reduce the efficiency of emergency services is that inclement weather conditions would reduce road capacity, increase transfer time, and sometimes block roads completely (Agarwal et al., 2006; Chang et al., 2013; Cools et al., 2010; Suarez et al., 2005; Zhang and Chen, 2019), which leads to the reduction of spatial accessibility and delay of response time. In addition, accidents such as traffic accidents and lightning accidents are more prone to occur in inclement weather, which increases the demand for EMSs (Edwards, 1996; Ramgopal et al., 2021). For example, on July 21, 2012, Beijing was hit by a rainstorm, with the average cumulative rainfall reaching 170 mm, caused 63 roads to be seriously flooded. This rainfall event led to a one-third increase in the number of calls to the emergency center, and the transfer time of ambulances was significantly prolonged, taking approximately 1.5~2 hours for each evacuation during the rainstorm. Usually, the transfer time wouldn't be more than 1 hour. (Wang et al., 2013; Beijing Evening,2012) On February 6, 2022, in Cleveland, US, an ambulance got stuck in the snow causing a long delay getting the patient to the hospital (Fox 8 News, 2022). On August 3, 2021, in Chattogram, Bangladesh, a daily rainfall of 190.6 mm caused many

ambulances with patients stuck in different areas of the city (Business Standard, 2021).
In the context of global climate change and rapid urbanization, extreme inclement
weather events strike cities more frequently (Huber and Gulledge, 2011; Stott, 2016;
Stott et al., 2016), the problem of urban rainstorms and waterlogging (the phenomenon
of a stagnant water disaster in an urban area due to heavy rainfall or continuous
precipitation) has become increasingly prominent. It is therefore of great importance to
investigate the influence of inclement weather on the spatial accessibility of EMSs.
The spatial accessibility of EMSs is defined by the travel impedance (distance or
time) between service locations and the scene (Guagliardo, 2004). A large body of
research on spatial accessibility is concerned with access to hospitals (Luo and Wang,
2003; Mao and Nekorchuk, 2013; Pan et al., 2018; Yang et al., 2020; Yin et al., 2021)
and first-aid stations (Hashtarkhani et al., 2020; Jones and Bentham, 1995; Shin and
Lee, 2018). To measure the EMSs accessibility, the two-step floating catchment area
(2SFCA) method is one of the common methods (Chen and Jia, 2019; Kanuganti et al.,
2016; Li et al., 2021; Luo and Qi, 2009). The 2SFCA method considers accessibility to
be mediated by not only the distance decay but also the interactions between supply
and demand (Chen and Jia, 2019), which is more suitable for normal scenarios. While
in the studies focusing on the influence of inclement weather on EMSs, people are more
concerned about the transportation situation, instead of the interaction between supply
and demand. The coverage analysis method (Coles et al., 2017; Green et al., 2017; Yu
et al., 2020) or shortest path analysis method (Albano et al., 2014; Andersson and
Stålhult, 2014) are more widely used. These methods could better characterize the
reduction of accessibility caused by the road service degradation. For example, Yu et
al. (2020) analyzed the accessibility of emergency service in England and identified
vulnerability hotspots by quantifying the EMSs coverage of area and population within
different time radii under different flood scenarios; Coles et al. (2017) measured the
travel time and service area coverage of EMSs in York, UK under flood scenarios by
using FloodMap-HydroInundation2D to model flood inundation; Yin et al. (2021)
assessed the vulnerability of EMSs to surface water flooding in Shanghai, China by
quantifying accessibility in terms of service area, population coverage and response
time. They simulated urban waterlogging scenarios under different rainfall intensities
and set traffic speed based on recorded average traffic speed under normal conditions,
which didn't consider the traffic speed variations induced by precipitation. Andersson
and Stålhult (2014) used network analysis methods to generate the shortest paths from
hospitals to various administrative areas in Manila, Philippines, and evaluated the
impact of different flood events on these paths. Most of these studies assumed that roads
are impassable or traffic speed has a certain degree of reduction when the flooded water
depth reaches a specific depth, and further evaluated the impact of rainstorm on EMSs
accessibility. Due to insufficient recorded traffic data, relatively few studies have been
performed to analyze the impact of road access capacity on EMSs accessibility
according to actual traffic speed variation.
In this study, we explore the impact of inclement weather on traffic and EMS
accessibility based on ground-truth traffic data. Beijing which is the capital of China is
used as a case study. The reductions in EMSs accessibility of Beijing under inclement
weathers in 2019 are quantified, and the urban-rural disparities in the distribution of
emergency medical facilities are further analyzed. Our study provides an approach for
evaluating the effectiveness and fairness of EMSs based on ground-truth traffic data,
and the results can not only provide reference for the optimization of EMSs in Beijing,
but also provide reference cases for other cities, which has a great practical significance.

## 2 Study area and dataset

### 2.1 Study area

Beijing, the capital of China, is located in the northern part of the North China Plain, with a total area of 16,410.54 square kilometers (Figure 1a). According to the seventh national census (National Bureau of Statistics, 2021), Beijing has a population of 21.89 million. As one of the largest metropolises in the world, Beijing has a monsoon-driven humid continental climate, with an average annual precipitation of approximately 600 mm, 80% of which is concentrated from June to September (Song et al., 2014). The terrain of Beijing is high in the northwest and low in the southeast, which is conducive to the formation of heavy rain and triggers strong convective weather. Beijing has a typical monocentric urban structure, and the area within the Six Ring Road is generally recognized as the urban core area. It is obvious that the density of transportation network and medical facilities in the urban area of Beijing are much higher than those in the suburbs (Figure 1b).

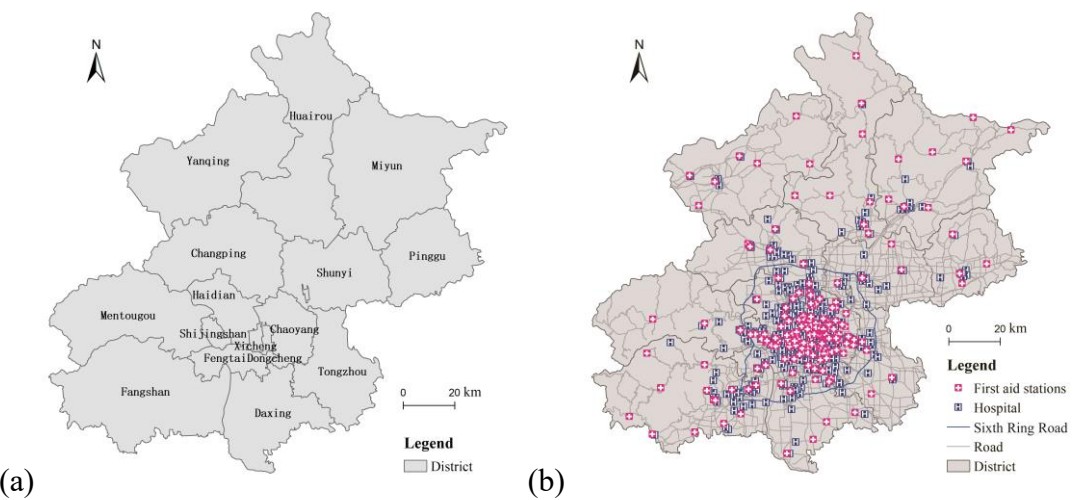

(a)                (b)

**Figure 1.** (a) Administrative division of Beijing and (b) EMS facility locations in Beijing, produced in ArcGIS 10.8.

**2.2 Dataset**

The data involved in this paper mainly include traffic data, meteorological data, EMSs data, and demographic data. Based on traffic data and meteorological data, we could build a topology road network (using node and edge primitives to describe interconnected linear features (roads) and points (roads junctions) on a map) with transfer time as impendence under inclement weather conditions and corresponding normal weather conditions. Combining the topology road network with medical facility locations and the distribution of the population by ArcGIS 10.8, we could further analyze the spatial accessibility of EMSs.

**2.2.1 Traffic and road network data**

The traffic data of Beijing are obtained from the Beijing Municipal Commission of Transport. The data span is from January 1, 2019, to December 31, 2019, including the average traffic speed (m/s) of each road section, updated every 2 min. The road network data contain 71,188 nodes and 81,523 edges, which can basically cover all the main roads in the whole Beijing area.

**2.2.2 Meteorological data**

The meteorological data utilized in this paper are TRMM (Tropical Rainfall Measuring Mission) precipitation data obtained from NASA, with a spatial resolution of 0.1°×0.1° (approximately 10 km×10 km) and a temporal resolution of 30 minutes. The whole city of Beijing is covered by 175 grids.

According to the classification of precipitation, moderate rain is defined as the rainfall is 5.0~14.9 mm per 12 hours (China Meteorological Administration, 2012). We chose intermediate value of the interval and average it to each hour. In this study, we

set a rule that if the precipitation of more than 10 grids (over 5% area of the city) in
Beijing is greater than 1.5 mm in 2 hours, it is considered a precipitation event. This
amount of precipitation may not high enough to cause the rainfall-runoff exceed the
drainage capacity of the sewer network in Beijing (DB11/ 685—2021, DB 11/T 1575—
2018). But the precipitation would cause slippery roads and decrease in drivers'
visibility, which would lead to a reduction of traffic efficiency and accessibility (Chu
and Fwa, 2018; Katz et al., 2012). The average precipitation of the whole city on each
date is averaged by the precipitation of all grids. In 2019, 19 working days of rainfall
and 3 working days of snowfall were selected.

**2.2.3 Medical facilities data**
The medical facilities mentioned in this paper mainly refer to two categories. One is
the first-aid stations, and the other is hospitals, as shown in Figure 1b. The locations of
these first-aid stations were obtained from the distribution map of first-aid stations
(Beijing Emergency Medical Center, 2021), including 72 stations in the downtown area
and 98 stations in the suburbs. The hospital point data were extracted from the online
map point of interest (POI) data of Beijing in 2019 (Gaode Maps, 2021). After
coordinate correction and deduplication, it contains a total of 630 general hospitals, 76
of which are third-level grade-A hospitals (the highest level in the evaluation system of
hospitals in mainland China).

**2.2.4 Demographic data**
The demographic data of 2019 were obtained from WorldPop (2018) with a spatial
resolution of 100 m×100 m. The data records present the population size.

## 3 Methodology

Figure 2 illustrates the methodology of this study. We first divide the weather conditions into two categories, inclement weather conditions and normal weather conditions, according to precipitation data. Second, the time impedance of each road section is analyzed based on the road network and traffic speed for both inclement and normal weather conditions, and the respective coverage rate of first-aid stations and the shortest transfer time to hospitals are calculated. Finally, the spatial accessibility to the population is calculated, and hotspots are identified. Both the service area analysis and the OD Cost Matrix analysis are GIS-based, and were done in ArcGIS 10.8. In this study, we made the following assumptions: (1) The ambulances move at the average speed all the time and would always take the shortest path in space; (2) In network analysis, the location of facilities is approximately considered to be on the nearest road point vertically; (3) In OD analysis, we use the centroid as the origin point to represent the whole grid, and the shortest path to hospital of all points within the grid is the same; (4) The prehospital EMSs is divided into two parts: the ambulances depart from the first-aid station to the scene and from the scene to the nearest hospital; (5) The proper limitation of EMS response time is 15 minutes. The case where patients transfer directly from the scene to an EMS facility via private transportation will not be considered in this study. (6) The hospitals' carrying capacity is not been considered in this study, and we assume that the demand of EMSs would not exceed the first aid stations' and hospitals' carrying capacity.

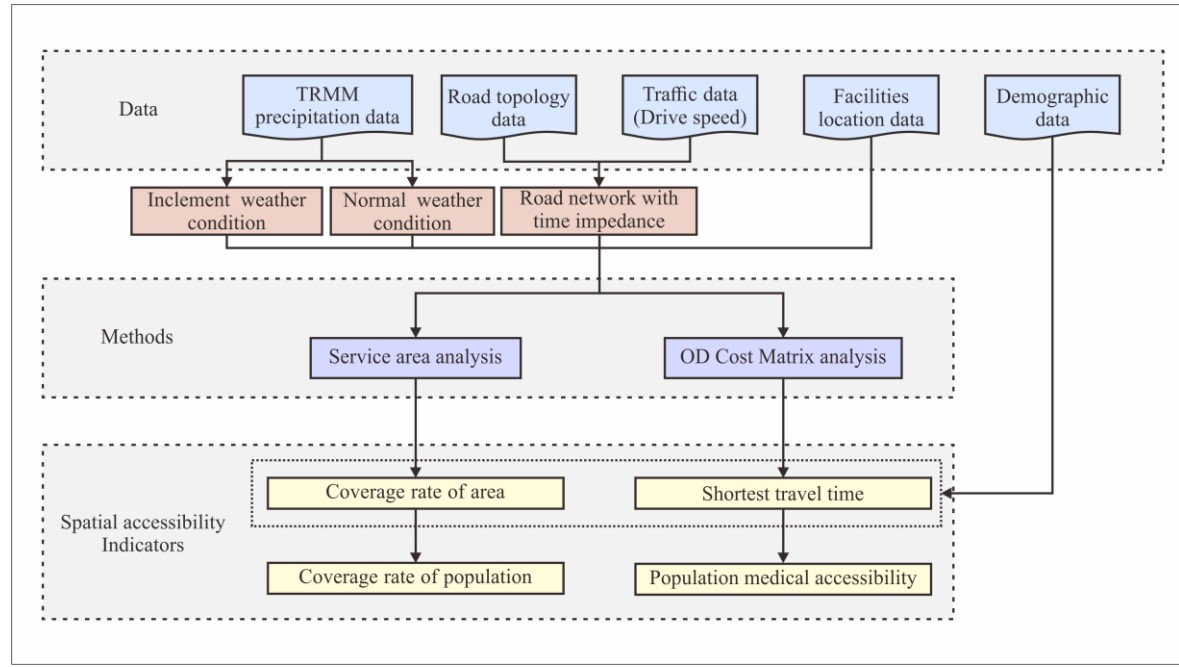


**Figure 2**. Methodology of this study, produced in CorelDraw 2019.


**3.1 Fluctuation of traffic speed under inclement weather**
For each weekday with precipitation, the traffic speed data of the selected period are
extracted and averaged. To avoid the inherent temporal variations of traffic speed
resulting from the day-of-week effects, holiday effects (Cools et al., 2007), season, and
other non-meteorological related factors, we introduce baseline days for inclement
weather days in this study to calculate the traffic speed fluctuation. For a given
precipitation day, we search for the same day of week in the two weeks forward and
backward to obtain the corresponding baseline days without precipitation. Only
nonholidays without precipitation events are selected as baseline days; otherwise, we
would continue to look forward or backward until four baseline days are found. The
average speed data of the four baseline days in the selected period were then averaged
as the baseline speed for the given precipitation day, and the traffic speed reduction rate
was calculated by eq. (1) and eq.(2):
$$r_c = \frac{v_p - v_b}{v_b} \qquad (1)$$
where $r_c$ is the traffic speed reduction rate in the selected period of the
precipitation day to its corresponding baseline day; $v_p$ is the traffic speed in the
selected period of the given precipitation day, and it is the average of the real-time
traffic speed in every 2 minutes during the selected time period in that day; $v_b$ is the
traffic speed in the selected period of the baseline precipitation days, which is calculated
by eq.(2):
$$v_b = \frac{\sum_{j=0}^{m} v_{d_j}}{m} \qquad (2)$$

where $v_{d_j}$ is the traffic speed in the selected period of a baseline day, and it is the
average of the real-time traffic speed in every 2 minutes during the selected time period
in that day; $m$ is the number of baseline days. In this case, $m$ equals 4. The average
traffic speed reduction rate is obtained by averaging the reduction rates of all roads with
reduced speed in the city.

**3.2 Analysis of coverage rate**
**3.2.1 The coverage rate of area**
A service area is a region that encompasses all roads that are accessible within a
specified impedance. Either distance or time can be used as impedance. In this study,
the time needed to pass through the road is calculated by the length of each road divided
by its corresponding traffic speed, and the service area analysis is carried out with time
as the impedance. In different scenarios, the time impedance varies, since the traffic
speed of each road is set according to the real-time traffic speed record of the chosen
date and chosen period. The core idea of the service area analysis function is to generate
service area polygons by setting each first-aid station as the starting point and the
traveling time as the driving radius. Under the inclement weather conditions and their
corresponding baseline conditions, the service area analysis of the 15-minute (Yin et
al., 2021) arrival time was carried out. The total area of the obtained service area
polygon is calculated to obtain the EMS coverage. The coverage rate of area is
calculated by eq. (3):
$$r_a = \frac{\sum A_s}{A} \times 100\% \qquad (3)$$

In eq. (3), $r_a$ is the coverage rate of the area; $A$ is the total area of the city, and $A_s$ is
the area of the service area.
**3.2.2 The coverage rate of population**
To analyze the matching degree between the EMS coverage and the population
distribution and identify the hotspots whose EMS coverage of the population is most
affected in inclement weather, we downscaled the calculation to the township scale.
Based on the grid population data of WorldPop and the coverage areas of EMSs under
different scenarios analyzed by service areas, we calculated the coverage rates of EMSs
of the population for each township. In each scenario, the polygon of service area
obtained from the result of service area analysis is used to mask the population grid,
and the covered population divided by the total population is the population coverage
of the township (eq. (4)).
$$r_p = \frac{\sum P_s}{P} \times 100\% \qquad (4)$$

In eq. (4), $r_p$ is the coverage rate of the population; $P$ is the total population of the
township, and $P_s$ is the population that is covered by the service area.

**3.3 The spatial accessibility to hospitals**
The spatial accessibility to hospitals is quantified by two indicators: the shortest
transfer time and the total transfer time. The shortest transfer time is calculated by the
OD (Origin-destination) cost matrix analysis method, which can find and measure the
minimum cost path from multiple starting points to single or multiple destinations in
the network. In this study, we calculate the minimum transfer time $od_i$ required for
each population grid centroid to reach the nearest hospital. To reduce the calculation
cost, the population grid data with 100 m resolution are aggregated and converted into
1000 m resolution. This could be interpreted as a sampling method, because we use the
centroid point of the grid to represent the other possible starting points in the grid, and
we ignored the tolerance caused by the travel time inside the grids.
The total transfer time is introduced to quantify the cumulative transfer time for each
population grid based on its population size, which is the number of potential users of
EMSs. The total transfer time is defined in this study by the shortest transfer time of
each population grid to the nearest hospital multiplied by its population. The numerical
value has no practical significance, and is only used for comparing the spatial
differences among regions. For each population grid centroid $i$, its total transfer time
($T$) is calculated by eq.(5):
$$T = od_i \times P_i \qquad (5)$$
In eq. (5), $od_i$ is the minimum transfer time, $P_i$ is the population of the grid.

**4 Results**
Based on the characteristics of morning and evening rush traffic flow on weekdays,
the diurnal variation in traffic can be divided into four periods: morning rush hours
(7:00-9:00), daily regular hours (9:00-17:00), evening rush hours (17:00-19:00), and
evening regular hours (19:00-22:00). We compared EMS coverage at different periods
of the day, and the results show that the period of morning rush hours has the most
significant negative impact on the accessibility of EMSs. We divided the city into the
inner city and suburban areas along the Sixth Ring Road. Taking the average 15-minute
coverage of the area of all Mondays in November as an example: (1) in the whole city
(both inner city and suburban), the coverage rate of EMSs is 38.72% in morning rush
hours, compared with 40% (±0.3%) in the remaining periods; (2) in the inner city, the
coverage rate is 77.37% in morning rush hours, compared with 83% (±0.6%) in the
remaining periods. Therefore, the accessibility of EMSs during the morning rush period
deserves more attention. Hence, our subsequent analysis is mainly concentrated on the
morning rush period.

**4.1 Impact of inclement weather on the traffic and EMSs coverage**
**4.1.1 The correlation between precipitation and traffic speed**
Figure 3 shows the relationship between average precipitation during morning rush
hours in the city and the average traffic speed reduction rate of all roads that have speed
loss in the city on weekdays. The unit of precipitation data is mm/2h, which indicates
the total precipitation in the 2 hours of morning rush hours. The negative values indicate
that the traffic speed decreases in inclement weather conditions. We could see that the
average traffic speed would decrease 10%~15% on most precipitation days. The
average speed decreases most on July 1st, July 9th, September 10th and December 16th,
reached 18%~25%. July 1st (Party's Day) and September 10th (Teachers Day) are special
days in China and the traffic speed is affected by both the inclement weather and traffic
control. December 16th was a snowy day with a precipitation of 0.13 mm/2h, and
snowfall has a greater impact on traffic than a rainfall with the same precipitation
(Agarwal et al., 2005). Figure 4 illustrates the spatial difference of traffic speed
reduction and distribution of precipitation on precipitation days. A large number of red
roads (with traffic speed reduction over 10 km/h) can be observed in the 4 days
mentioned above. By comparing the distribution of precipitation and traffic speed
reduction on different dates in Figure 4, it can be found that the precipitation in the four
days with the most severe speed reduction was moderate, and the precipitation
distribution of the whole city was relatively uniform. Compared with other rain days,
although the precipitation on July 5, August 9 and September 19 was larger and
concentrated in the inner city, the traffic speed reduction of the whole city was not as
serious as the four days mentioned above, which may be caused by the decrease of
people's willingness to travel with the increase of rain.

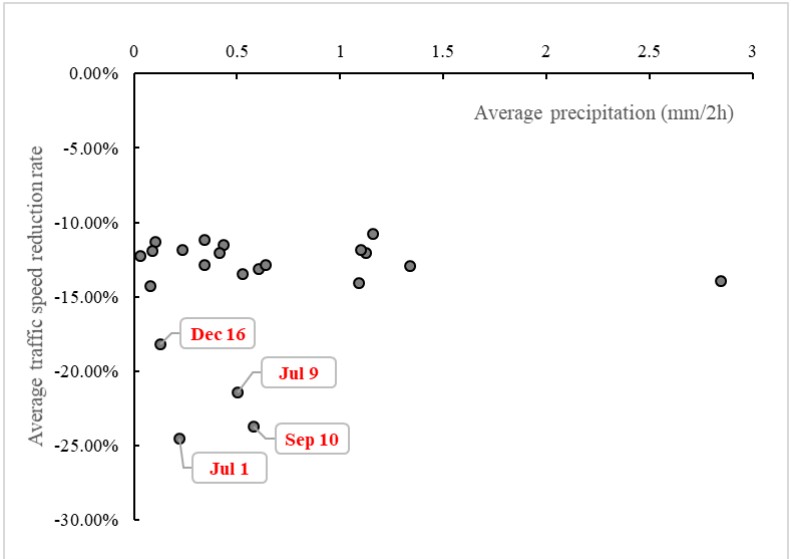


**Figure 3.** The correlation between average precipitation and average traffic speed reduction rate,
produced in Excel 2016.

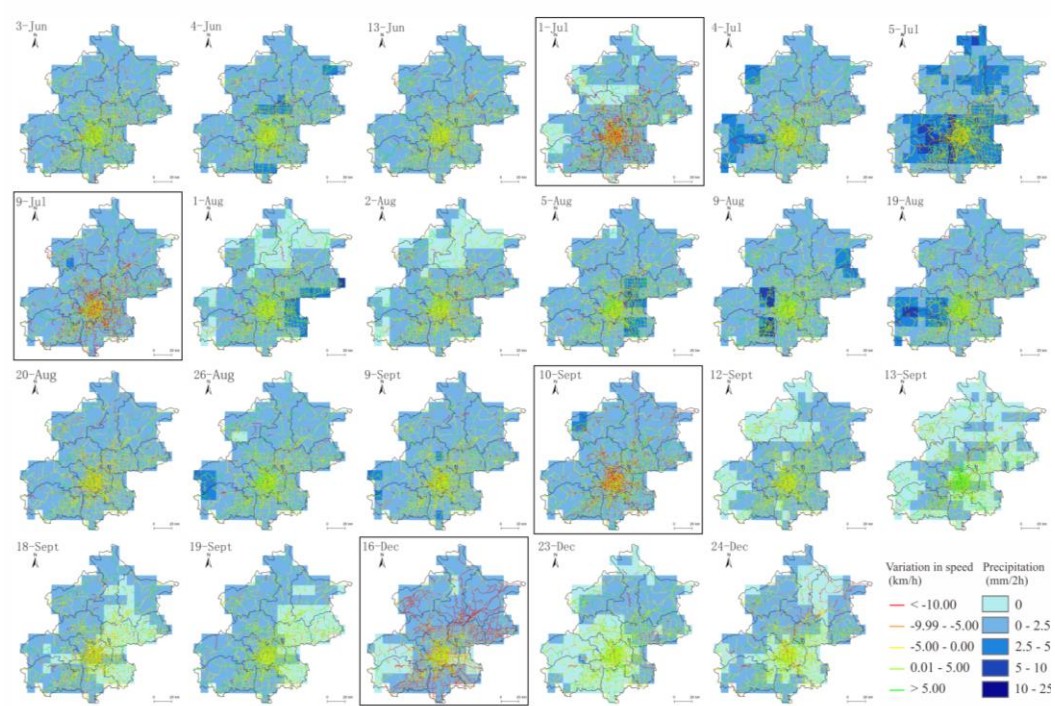


**Figure 4.** Variation in drive speed and distribution of precipitation on selected precipitation days
(the 4 subfigures with black borders shows the 4 most affected scenarios), produced in ArcGIS 10.8
and CorelDraw 2019.
**4.1.2 The correlation between precipitation and EMSs coverage rate**
The change in the coverage rate of EMSs was calculated by subtracting the coverage
rate under the inclement weather condition from that under the corresponding baseline
condition. Figure 5 shows the correlation between the average precipitation during
morning rush hours and the relative change values of the EMS coverage rate of the area.
The negative values indicate that the coverage of EMSs decreases in inclement weather
conditions. Consistent with the pattern of the traffic speed reduction, the worst loss of
coverage rate also occurred on three rainy days: 1$^{st}$ July (Mon), 9$^{th}$ July (Tue), and 10$^{th}$
September (Tue), and one snowy day: 16$^{th}$ December (Mon), in which the 15-minute
EMS coverage rate reduced by 4.6%, 5.6%, 4.2% and 13.3%. Combined with the spatial
distribution of precipitation and traffic variation (Figure 4), the snowfall on December
16$^{th}$ caused a large traffic speed reduction of the suburban roads, which led to a
significant reduction in overall EMS coverage. Therefore, we chose these four days as
the worst weather scenario of the year and analyzed the spatial differences of medical
accessibility in the whole city.

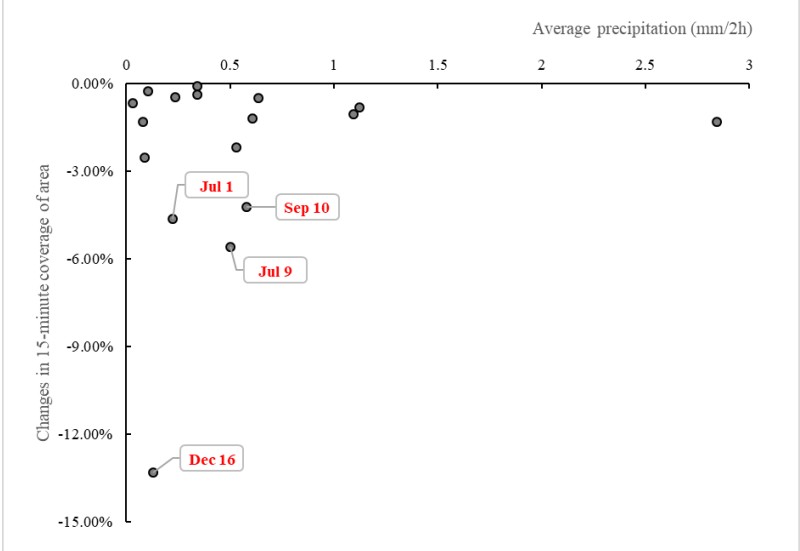

**Figure 5**. The correlation between the average precipitation and the relative change of the EMS
coverage rate of the area, produced in Excel 2016.

**4.2 The spatial distribution of EMS accessibility under the worst scenario**

**4.2.1 EMSs coverage rate of population**

We calculated the 15-minute EMS coverage rate of the population under the four
most severely affected inclement weather conditions of 1st July, 9th July, 10th September,
and 16th December and their corresponding baseline conditions at the township scale in
Beijing. Figure 6 shows the 15-minute EMSs coverage rate of population under four
most severely affected inclement weather conditions of 1st July, 9th July, 10th
September and 16th December and their corresponding baseline conditions at the
township scale in Beijing. The results demonstrate that most parts of downtown areas,
including Dongcheng District, Xicheng District, Haidian District, and Chaoyang
District, could have 90%–100% population coverage of EMSs, regardless of the
weather conditions. In the large area of suburbs, the coverage rate of the population
varied from lower than 30% to 90%. Under inclement weather conditions, the coverage
rate in some towns in the suburbs would drop sharply, with the worst townships having
a 40% reduction. The reason behind this difference is that the distribution of first-aid
stations in Beijing is similar to the distribution of the road network, which is dense in
the central urban area and sparse in the suburbs.

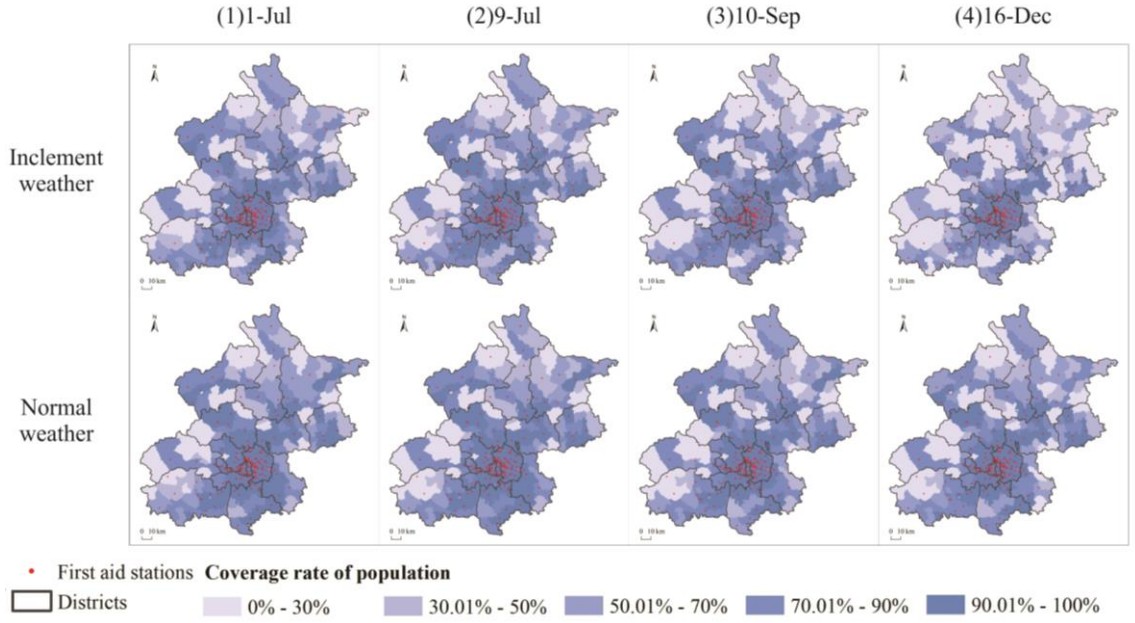


**Figure 6.** The EMSs coverage rate of population in townships under the inclement weather condition
and normal weather condition on 1st July, 9th July, 10th September and 16th December, produced in
ArcGIS 10.8 and CorelDraw 2019.

To illustrate the impact of inclement weather on the EMS coverage rate of the
population more clearly, Figure 7 shows the change in the EMS coverage rate of the
population in townships in inclement weather relative to normal weather on the four
days. The results identify several townships in the outer suburbs (Miyun, Huairou,
Pinggu and Yanqing districts) that would experience the most severe decrease in
population coverage under inclement weather conditions, with a maximum reduction
of more than 40%. These areas are hotspots that need to draw attention in EMS
construction planning. The suburb areas, such as Shunyi, Daxing, and Tongzhou, are
more vulnerable to inclement weather as they have less distribution of medical facilities
and sparser road networks, as well have a relatively higher proportion of the elderly
population over the age of 80. The average proportion of the elderly is 1.88% in the
whole city, 1.37% in the inner suburbs and 2.04% in the outer suburbs. On December
16th, 12.6% of population (3.5 million) could not get EMS within 15 minutes, compared
to 7.5% with the baseline condition.

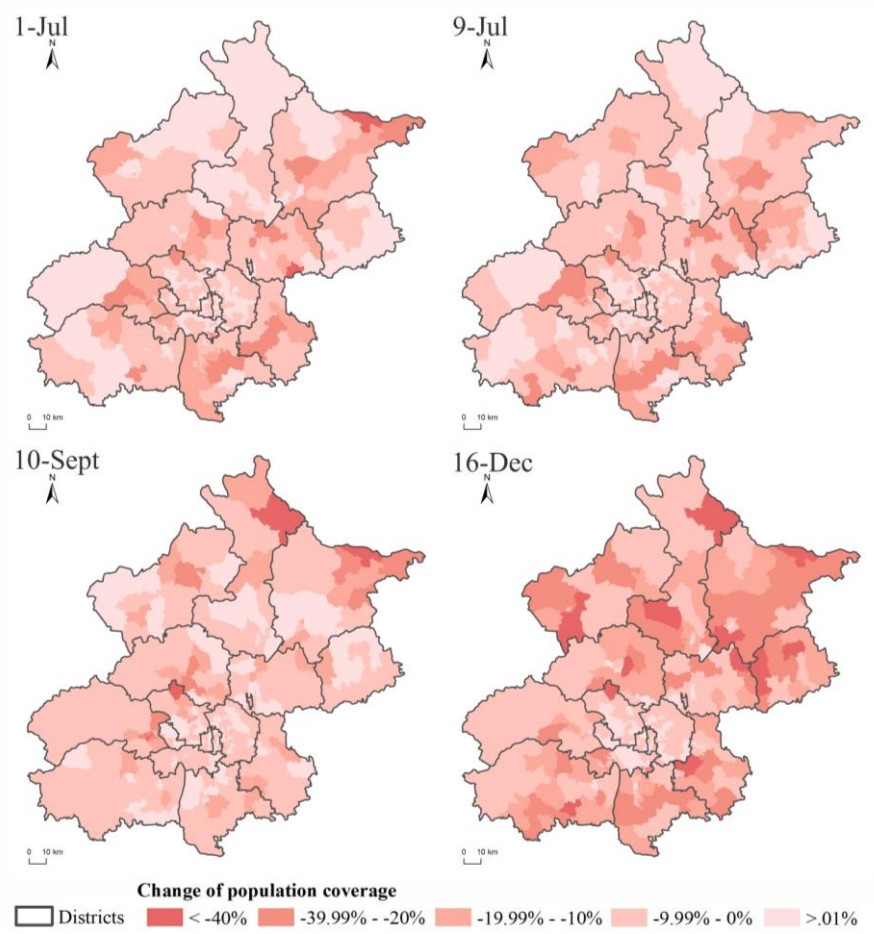


**Figure 7.** The change in EMS coverage rate of the population in townships in inclement weather
relative to normal weather on 1st July, 9th July, 10th September, and 16th December, produced in ArcGIS
10.8 and CorelDraw 2019.
Figure A1 shows the correlation between the baseline EMS coverage rate of the
population of each township and its reduction under inclement weather. The results
reveal that the population of the towns with low baseline EMS coverage rate would lose
more EMS coverage under inclement weather, especially on snowy day. The average
traffic speed reduction in the urban area (within the Sixth Ring road) was -26.64%, -
23.27%, -25.20% and -15.77% on 1st July, 9th July, 10th September, and 16th
December, while that in the suburban area (outside the Sixth Ring Road) was -19.59%,
-19.08%, -17.27% and -23.21%. Based on the results, we analyzed the reasons why that
suburban area would become more vulnerable under inclement weather. Combined
with the traffic speed reduction and the EMS coverage reduction, on rainy days,
although the urban area has more traffic speed loss, the suburban area still experiences
more EMS coverage loss. Once the inclement weather affects the traffic on some road,
the urban areas still have many other roads than can bypass, but not in suburbs. On
snowy days, the suburban area has more traffic speed reduction, and with the sparser
road network, the EMS coverage in the suburban area would shrink much more than
rainy days.

**4.2.2 The accessibility to hospitals**
Figure 8 shows the increased transfer time from each population grid to the nearest
hospital under the four inclement weather conditions of 1st July, 9th July, 10th September,
and 16th December relative to the baseline condition. The value indicates the impact of
inclement weather on accessibility to hospitals. The situation is slightly different on
rainy days and snowy days. On rainy days, the shortest time to reach the nearest hospital
generally could increase by 0–10 minutes in most parts of Beijing due to slower traffic
speed on the roads caused by rain. Although in some small parts of suburban areas, the
shortest time to the nearest hospital would be slightly shortened on indicating that the
traffic will be smoother in some areas when it rains, which may be due to the reduction

of traffic demand (Maze et al., 2006). While on 16[th] December, affected by snow, the whole city's road traffic generally slowed down, and the transfer time to the nearest hospital increased by 10–40 minutes. The western part of Mentougou District and a small part of the northern Yanqing District were the most affected, with the time needed to reach the nearest hospital prolonged by more than 30 minutes, up to 45 minutes. In Huairou district, the eastern part of Yanqing district, and the northern part of Miyun district, the transfer time was also prolonged by 11–30 minutes.

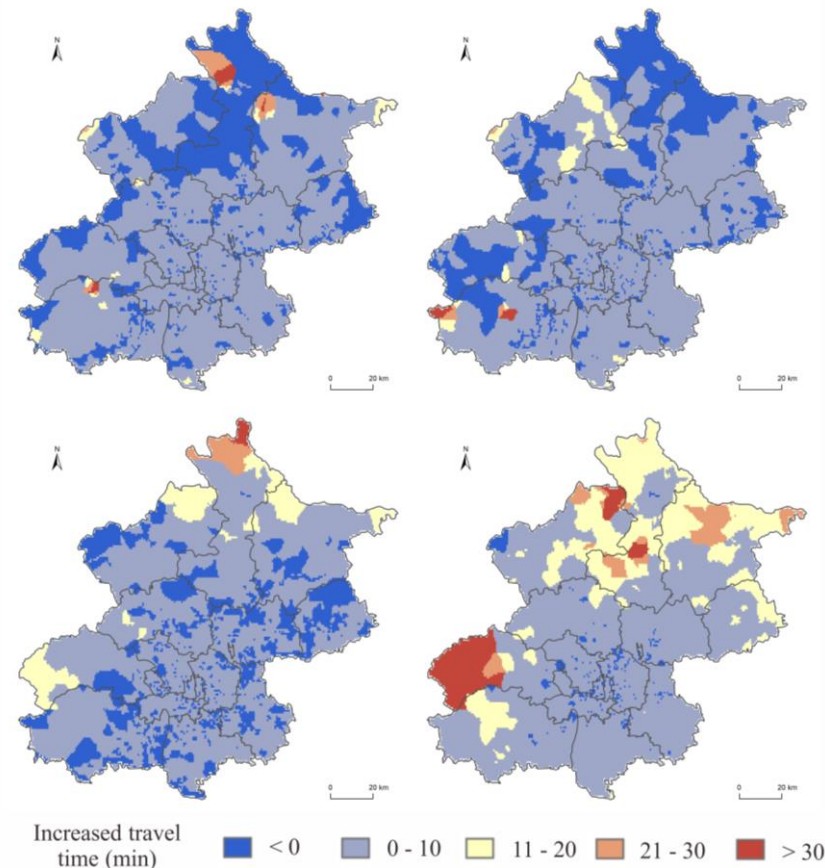

**Figure 8**. Increased transfer time to hospitals on 1[st] July, 9[th] July, 10[th] September, and 16[th] December, produced in ArcGIS 10.8 and CorelDraw 2019.

We did a zonal statistic of the average baseline transfer time to hospital and the average increased transfer time to hospitals to each town, and the correlation between the two indicators shown in Figure A2 indicate the similar pattern with the EMS

coverage, which is the towns with low baseline accessibility to hospitals would also
more affected by inclement weather.
Overlaying the demographic grid data, the size of the population affected by a
delay of over 10 minutes would be 0.02 million on 1st July, 0.03 million on 9th July,
0.05 million on 10th September, and 0.3 million on 16th December.
Figure 9 shows the change in the total transfer time under inclement weather
conditions on 1st July, 9th July, 10th September, and 16th December, relative to the
baseline conditions. The results show that on three rainy days, 1st July, 9th July, and 10th
September, within the Sixth Ring Road extent, the total transfer time increased
significantly under inclement weather, which means that, although the transfer time
would not increase much in urban areas, due to the high population density, the
cumulative delay time for total potential demand would be significant. In the suburbs,
the total transfer time would increase slightly or even decrease, especially in some areas
of Huairou, Yanqing, and Miyun districts, which means that, although the transfer time
would increase greatly, due to its low population density, the cumulative delay time for
total potential demand would not be serious. However, due to the influence of snowfall
on 16th December, the total transfer time in the whole city was slightly or moderately
increased, and there were almost no regions where the total transfer time decreased,
which means snowfall would cause an even cumulation of delay time for total potential
demand across the whole city, both urban and suburban.

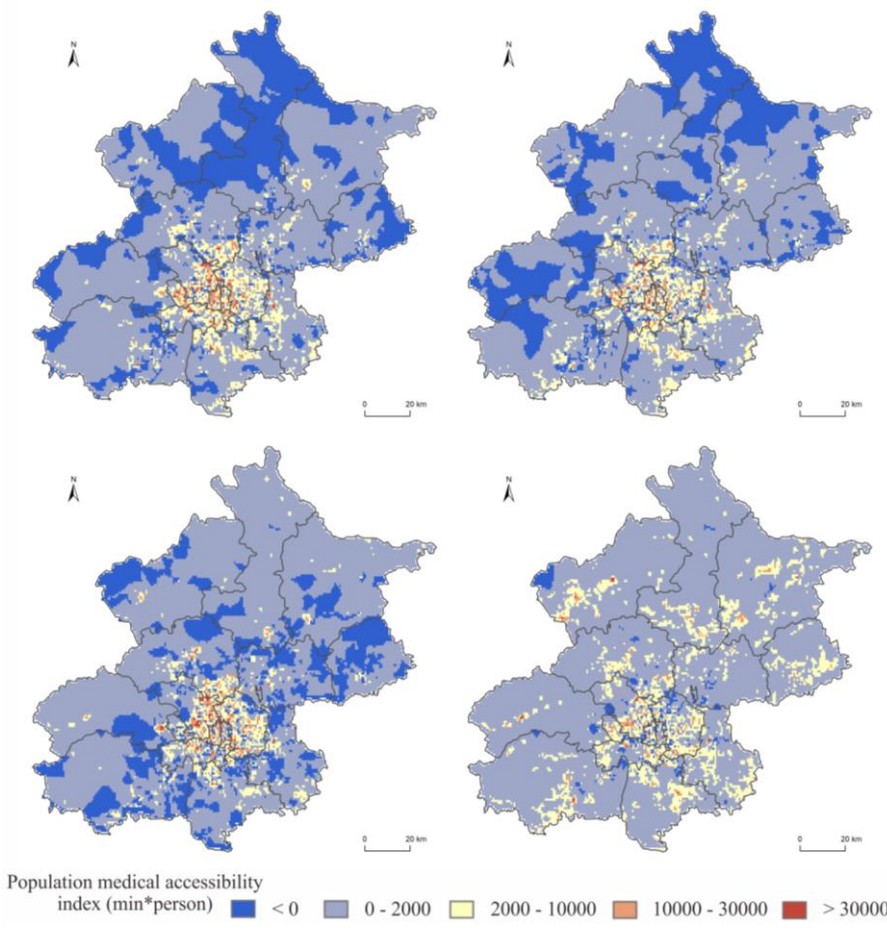

Population medical accessibility index (min*person) < 0  0 - 2000  2000 - 10000  10000 - 30000  > 30000

**Figure 9**. The change in the total transfer time on 1st July, 9th July, 10th September, and 16th December, produced in ArcGIS 10.8 and CorelDraw 2019.

## 5 Conclusions and discussion

Our study evaluates the spatial accessibility of EMSs in Beijing under different weather conditions in 2019 based on city-scale ground-truth traffic data updated every 2 minutes. The spatial accessibility of EMSs was quantified by the coverage rate of the first-aid stations' service area, the coverage rate of first-aid stations' service population, and the shortest transfer time to the nearest hospital. Our study reveals the influence of precipitation on the accessibility and equity of EMSs, which could help guide EMS construction planning in cities, get prepared for extreme weather conditions, and finally assist the decision-making of the corresponding government departments. The main

conclusions are as follows:
First, the results show that inclement weather, such as rainfall and snowfall, could
have a negative impact on the accessibility of EMSs overall. Precipitation reduces the
driving speed of vehicles on the road, thus reducing EMS coverage. In severe cases, the
EMS coverage rate of the area can be reduced by more than 10%. Besides, snowfall has
a greater impact on EMSs accessibility than rainfall.
Second, the EMSs accessibility is more affected by inclement weather in places with
low baseline accessibility to EMSs. And the results reveal a serious rural-urban
disparity in emergency medical facilities distribution in Beijing: The EMSs
accessibility of population in some townships of the outer suburbs is very low and
would also greatly reduce under inclement weather.
Third, some specific days may affect the traffic flow, which has an amplification
effect on the traffic congestion caused by inclement weather. When they encounter the
inclement weather, there are potential risks of decrease of traffic efficiency and EMSs
accessibility, which should be given sufficient attention.
To the best of our knowledge, there was no studies have been performed to analyze
the impact of road access capacity on EMSs accessibility under inclement weather
according to actual traffic speed variation. Our study provides an attempt to analyze the
spatial accessibility of EMSs under inclement weather based on city-scale ground truth
traffic data and meteorological data, where the former is usually difficult to obtain. In
previous literatures (Yin et al., 2021; Coles et al., 2017; Albano et al., 2014), simulation
methods were widely used on the research on EMSs accessibility or traffic capacity
under inclement weather; however, the ground-truth traffic data that covers every road
in the whole city under precipitation scenarios, was hardly used in the previous studies
of the impact of weather on traffic and accessibility. Our study could be a good
empirical verification in this field of study. We also found that snowfall may have a
greater impact, which is hard to find out using flood simulation methods. The results
from this study provide a scientific reference for city planning departments in Beijing
to optimize the site selection of emergency service facilities and get prepared for traffic
dispersion on inclement weather. The relevant methods mentioned in this paper are also
suitable for both holidays and workdays and can be easily applied to other cities once
traffic data or empirical formulas regarding the impact of inclement weather on road
traffic can be obtained.
There are also some parts in our research that can be improved in future research.
First, we averaged the traffic speed reduction rate of all the roads in the city, as well as
the precipitation data, which could conceal congestion hotspots. Besides, all the
calculation was done by the 2-hour selected period, which may neglect the delayed
responses of rainfall runoff and temporal variation of rainfall. In further studies, with
higher resolution precipitation, along with corresponding traffic data, we could narrow
the scale to blocks, pay more attention to local congestions, and analyze the correlation
of precipitation and traffic speed on a finer scale. Second, due to the data limitation, we
could only analyze the EMSs accessibility in 2019, and the precipitation intensity in
this year was not quite high. If had longer time series precipitation and traffic data, we
could analyze the impact of precipitation magnitude to the traffic and accessibility,
instead of simply dividing the days in a binary manner into inclement and non-
inclement weather days. Under such precipitation conditions, the EMSs accessibility
has been affected to a certain extent, and it would be much more difficult to get timely
EMSs under even more extreme inclement weather condition. In previous studies, Yin
et al. (2021) found that surface water flooding could result in nonlinear impacts on EMS
spatial coverage in Shanghai: 5- and 20-year pluvial flooding both exerted very minor
and the effect of 100-year surface water flooding appears to be more pronounced.
Future studies should take extreme precipitation events into account. Third, due to the
lack of high-resolution DSM (Digital Surface Model) data, we did not run a
hydrological flood simulation in Beijing, which could reveal the relationship of
precipitation and the actual amount of water on the streets. This could be improved in
the future studies with more high-resolution topographic data. Fourth, we used the "15-
minutes arrival time" as a main boundary in this study, however, the proper response
time would vary in different countries or cities. So, the setting of response time
boundary should be adjusted considering the actual situation of the city when the
method in this paper is applied to other cities. Fifth, we aggregated the population grid
evenly in the city. If a varying resolution could have been applied with a finer grid in
the heavily populated center, and a coarser grid towards the outskirts, it may capture
more of the dynamics in a metropolis with varying population and infrastructure
densities.
**Appendices**
**Appendix A**

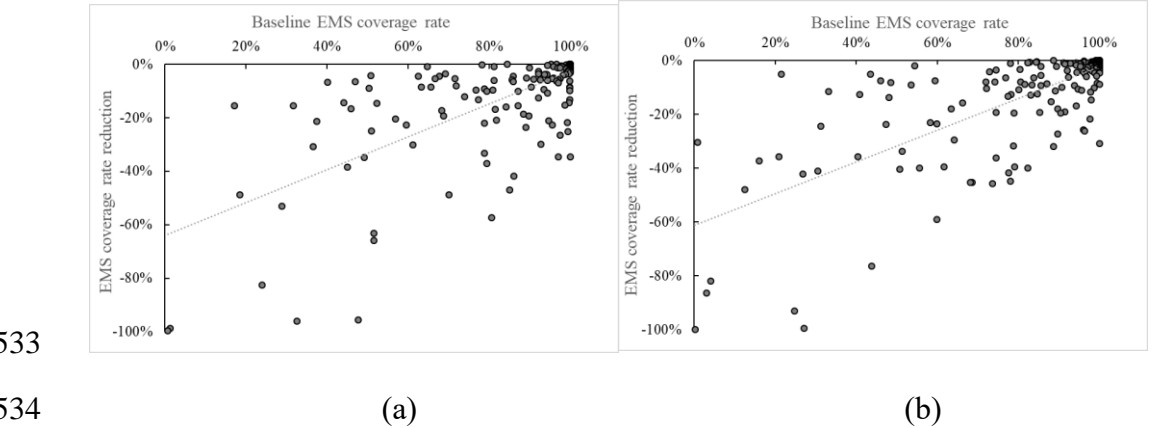

534                 (a)                        (b)

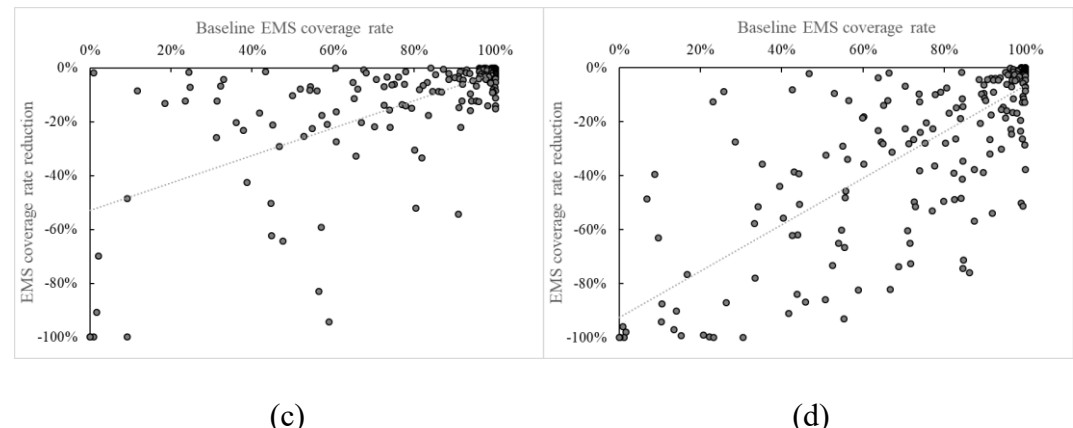


536                                  (c)                                              (d)

**Figure A1.** The correlation between the baseline EMS coverage rate of population and its reduction
percentage in inclement weather. (a) 1st July, (b) 9th July, (c) 10th September, and (d) 16th December,

539                                    produced in Excel 2016.


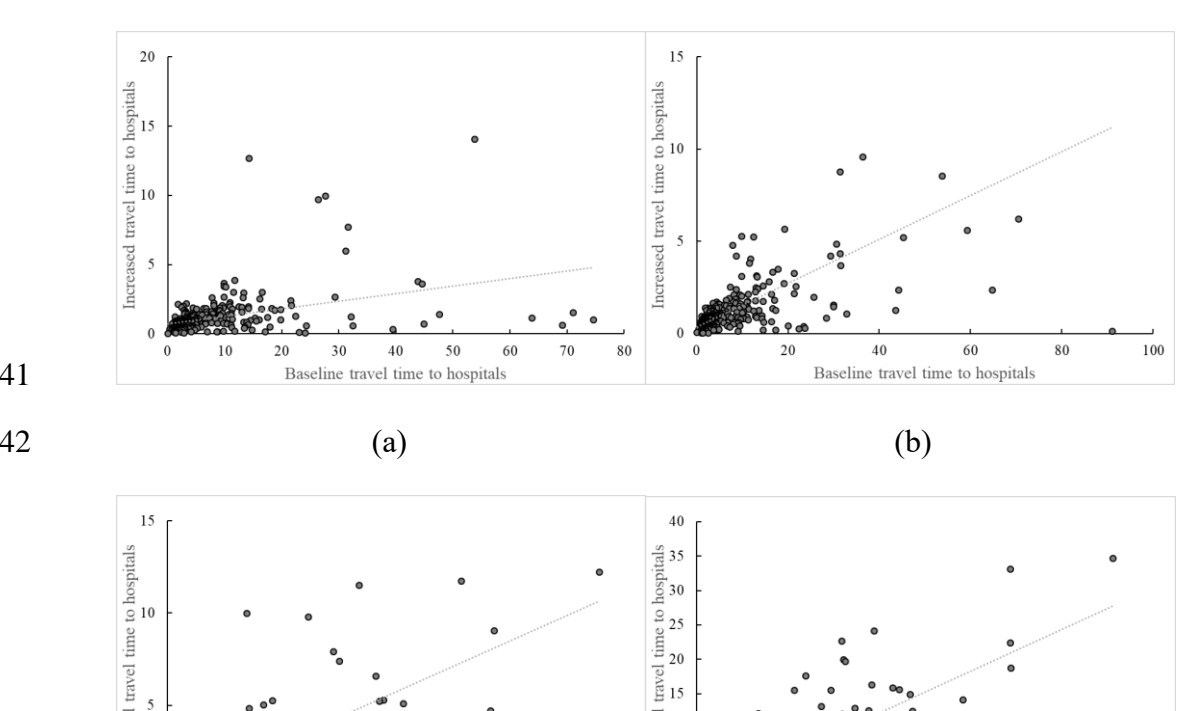


542                                  (a)                                              (b)


544                                  (c)                                              (d)

**Figure A2.** The correlation between the baseline transfer time to hospitals and the increased transfer
time in inclement weather. (a) 1st July, (b) 9th July, (c) 10th September, and (d) 16th December, produced

547                                      in Excel 2016.


## Data availability

All raw data can be provided by the corresponding authors upon request.

## Author contributions

KL planned the research; JZ, ML provided the traffic data; YZ and KL analyzed the
data, YZ wrote the manuscript draft; KL, XN, MW, and DY reviewed and edited the
manuscript.

## Competing interests

The authors declare that they have no conflict of interest.

## Acknowledgments

The study is supported by the Major Program of National Natural Science
Foundation of China (No. 72091512) and National Natural Science Foundation of
China (41771538). The financial support is highly appreciated.

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
