# Peer review of "Spatial accessibility of emergency medical services under inclement"

_Natural Hazards and Earth System Sciences, 2022_

## Author Comment (AC1)

**Referee #1**

The study looks at the spatial accessibility of emergency medical service facilities in Beijing resolved according to weather situations, days and time-of-the-day. The authors demonstrate an empirical approach for linking spatially resolved accessibility decreases to weather situations, and manage to point out spatial inequalities on a sub-urban level. The approach, despite the reviewer having some reservations regarding certain over-simplifications along the way, shows a way forward to combine empirical data to yield insights into an under-studied topic (EMS accessibility) while shedding light on a dimension of social inequity.

**General**

1. L 28 not clear at that stage what is meant with « coverage rate ». Is it the total area covered within a 15 mins response time? i.e. 13 % reduction refers to a km2 number? why then « rate » (I do know that it is explained later on in the text, however the abstract should be understandable before this)?

**Response**: We thank for reviewer's comment. Using "coverage rate" in the abstract may cause confusion to readers, in the revised manuscript, we have rephrased the sentence in Lines 27-29, Page 2:

*Under inclement weather scenario, the area in the citywide that could get EMSs within 15 minutes would decreased by 13% compared to normal scenario. And in some suburban townships, the population that could get 15-min EMS would decrease by 40%.*

2. L 45-55: It is good that the authors give detailed insight into what parts make up EMS, especially tailored to the case study context. However, there seems to be - strictly speaking - some inconsistency in the use of the term services: One case includes the transport to the EMS facilities within the service definition (aka, the transportation service), the other case not (only treatment service).

Further, please elaborate how this goes together with the definition in L 72-74: It seems that this definition covers only the case of responders starting out from an EMS facility, getting to the scene, and transferring back to a respective facility (e.g. via ambulance). From the initial description above, the reader might think the authors will cover, however, also the case where patients transfer directly from the scene to an EMS facility (e.g. via private transportation). Hence, it might be good to explicitly mention again that this latter case will not be covered, even though described above for reasons of completeness.

**Response**: We thank for reviewer's comment. The EMSs includes both pre-hospital and in-hospital services, and in our study, we focus on the accessibility of EMSs (only the transportation part). And we divide the process into 2 parts: one is starting out from an EMS facility and getting to the scene (as in 3.2), the other is transferring the patient from the scene to hospitals (via ambulance, not private transportation, as in 3.3). So, we assume that the patients get the EMSs when the ambulance arrives

the scene, and the EMSs is complete when the ambulance transfers the patient to hospital, and both parts were covered in our study. The case where patients transfer directly from the scene to an EMS facility via private transportation was not covered in this study. We apologize for the unclarity, and we mentioned it in the revised manuscript in Lines 193-195, Page 9:

*The case where patients transfer directly from the scene to an EMS facility via private transportation will not be considered in this study.*

3. L 58: To the non-local reader, it is unclear whether 1.5 to 2 hours response time is significantly longer than normally. One would assume this, but it would be helpful to provide average response times during normal conditions for comparison.

**Response**: We thank for reviewer's comment. As suggested by the reviewer, we have added the comparison of normal condition in Lines 61-62, Page 3. And this information comes from the same news.

*while the transfer time wouldn't be more than 1 hour on usual*

4. L 78: Reference to the 2SFCA method?

**Response**: We thank for reviewer's comment. We have added the reference to 2SFCA methods in Lines 79-80, Page 4:

*Chen, X., and Jia, P.: A comparative analysis of accessibility measures by the two-step floating catchment area (2sfca) method, INT J GEOGR INF SCI, 33, 1739-1758,2019.*
*Kanuganti, S., Sarkar, A. K., and Singh, A. P.: Quantifying accessibility to health care using two-step floating catchment area method (2sfca): a case study in rajasthan, Transportation Research Procedia, 17, 391-399,2016.*
*Li, M., Kwan, M., Chen, J., Wang, J., Yin, J., and Yu, D.: Measuring emergency medical service (ems) accessibility with the effect of city dynamics in a 100-year pluvial flood scenario, CITIES, 117, http://doi.org/10.1016/j.cities.2021.103314, 2021b.*
*Luo, W., and Qi, Y.: An enhanced two-step floating catchment area (e2sfca) method for measuring spatial accessibility to primary care physicians, HEALTH PLACE, 15, 1100-1107,2009.*

5. L 74-L104: it is good that you provide some literature revolving around the topic of study. However, this very brief listing-style does not make it very clear how those papers are related to the concrete problem statement, or not summarized into coherent areas of challenges, seeming a bit randomly aggregated.

**Response**: We thank for reviewer's comment. We recomposed the whole part of review in the revised manuscript, please refer to Lines 73-105, Pages 4-5:

*The spatial accessibility of EMSs is defined by the travel impedance (distance or time) between*

*service locations and the scene (Guagliardo, 2004). A large body of research on spatial accessibility is concerned with access to hospitals (Luo and Wang, 2003; Mao and Nekorchuk, 2013; Pan et al., 2018; Yang et al., 2020; Yin et al., 2021) and first-aid stations (Hashtarkhani et al., 2020; Jones and Bentham, 1995; Shin and Lee, 2018). To measure the EMSs accessibility, the two-step floating catchment area (2SFCA) method is one of the common methods (Chen and Jia, 2019; Kanuganti et al., 2016; Li et al., 2021; Luo and Qi, 2009). The 2SFCA method considers accessibility to be mediated by not only the distance decay but also the interactions between supply and demand (Chen and Jia, 2019), which is more suitable for normal scenarios. While for studies focusing on the influence of inclement weather on EMSs, people concern more about the transportation situation, instead of the interaction between supply and demand. The coverage analysis method (Coles et al., 2017; Green et al., 2017; Yu et al., 2020) or shortest path analysis method(Albano et al., 2014; Andersson and Stålhult, 2014) are more widely used. These methods could better characterize the reduction of accessibility caused by the road service degradation. For example, Yu et al. (2020) analyzed the accessibility of emergency service in England and identified vulnerability hotspots by quantifying the EMSs coverage of area and population within different time radii under different flood scenarios; Coles et al. (2017) measured the travel time and service area coverage of EMSs in York, UK under flood scenarios by using FloodMap-HydroInundation2D to model flood inundation; Yin et al. (2021) assessed the vulnerability of EMSs to surface water flooding in Shanghai, China by quantifying accessibility in terms of service area, response time, and population coverage, considering four temporal scenarios (nighttime, evening, daytime, and morning–evening peak) of average drive speeds based on a real-time traffic analysis from GPS big data; Andersson and Stålhult (2014) used network analysis methods to generate the shortest paths from hospitals to various administrative areas in Manila, Philippines, and evaluated the impact of different flood events on these paths. Most of these studies assumed that roads are impassable or traffic speed has a certain degree of reduction when the flooded water depth reaches a specific depth, and further evaluated the impact of rainstorm on EMSs accessibility. Due to insufficient recorded traffic data, relatively few studies have been performed to analyze the impact of road access capacity on EMSs accessibility according to actual traffic speed variation.*

6. Section 3.1:

Could you please elaborate the reason for averaging to daily speeds for the baseline constructions, since you later also look at rush-hours and non-rush-hours specifically?

In a similar line of argumentation, averaging hence-obtained speed reduction rates across all road sections within the city (L220f), seems to conceal congestion hotspots? Please elaborate why this was done and the potential limitations of this.

Also, it is not clear to me from this description, if days are simply divided in a binary manner into inclement and non-inclement weather days, irrespective of the precipitation magnitude? Please elaborate in more detail if this is the case, and what was the reasoning for and potential shortcoming of this.

**Response**: We thank for reviewer's comment. Actually, we averaged the traffic speed data of the

selected period, not of the whole day (as in line 219 and line 221). We apologize for the unclarity, and we added "in the selected period" in Lines 207-215, Pages 10:

*The average speed data of the four baseline days **in the selected period** were then averaged as the baseline speed for the given precipitation day, and the traffic speed reduction rate was calculated by eq. (1):*

$$r_c = \frac{v_p - \frac{\sum_{j=0}^{m} v_{d_j}}{m}}{\frac{\sum_{j=0}^{m} v_{d_j}}{m}} \qquad (1)$$

*where $r_c$ is the traffic speed reduction rate **in the selected period** of the precipitation day to its corresponding baseline days; $v_p$ is the traffic speed **in the selected period** of the given precipitation day; $v_{d_j}$ is the traffic speed **in the selected period** of a baseline day, and m is the number of baseline days. In this case, m equals 4.*

We admit that averaging hence-obtained speed reduction rates across all road sections within the city do conceal the congestion hotspots, which is a limitation of this study. This limitation could be improved in the further study on a small scale based on a higher resolution of precipitation. Nevertheless, averaging could give us the overall impact of precipitation on vehicle speed across the city. And we put Figure 4 to complement the spatially difference and find out the congestion hotspots. We also admit this limitation in Lines 487-492, Pages 25:

*However, there are also some limitations in this study. First, we averaged the traffic speed reduction rate of all the roads in the city, as well as the precipitation data, which could conceal congestion hotspots. In further studies, with higher resolution precipitation, along with corresponding traffic data, we could narrow the scale to blocks, pay more attention to local congestions, and analyze the correlation of precipitation and traffic speed on a finer scale.*

In this study the days were simply divided in a binary manner into inclement and non-inclement weather days, the reason is that we only got the traffic data of 2019, and there are few days with precipitation. The data does not support an analysis considering precipitation magnitude. In the revised manuscript, we have acknowledged this shortcoming in the discussion section in lines 499-492, page 25:

*Due to the data limitation, we could only analyze the EMSs accessibility in 2019, and the precipitation intensity in this year was not quite high. If with more precipitation and traffic data, we could analysis the impact of precipitation magnitude to the traffic and accessibility, instead of divided the days in a binary manner into inclement and non-inclement weather days.*

7. Section 3.3: How does aggregation of the population grids to 1000m in a city distort potential travel patterns? Given that the topology of the road network within Beijing is at a much higher resolution, does this aggregation not lead to a very coarse estimation of what roads are taken, and which ones not?

**Response**: We thank for reviewer's comment. When we run the OD cost matrix analysis for each

population grid, we use its centroid as the origin point, not the whole square, so it won't affect the resolution of the topology of the road network. This might be interpreted as a sampling survey. The choice of roads to the nearest hospital won't be greatly impact because the distance to the nearest to the nearest hospital is normally more than 1km in most areas. The reason of aggregation the population grids are mainly reducing the amount calculation, because the OD cost matrix analysis is computationally intensive. We added some explanation in Lines 256-258, Page 12:

*This could be interpreted as a sampling method, because we use the centroid point of the grid to represent the other possible starting points in the grid, and we ignored the tolerance caused by the travel time inside the grids.*

8. Section 4. Results:

L275-280: It is hard to understand to which scenarios / analysis areas the percentages belong. Do 38 and 40%, resp., refer to the city including suburban areas? And 77 and 83% refer to only the inner city (is that meant by Six Road Ring)? Please phrase it in a way that describes the area analyzed better to a non-local.

**Response**: We thank for reviewer's comment. The reviewer's comprehension is correct. And we revised the sentences in lines 273-279, page 13:

*We divided the city into the inner city and suburban areas along the Sixth Ring Road. Taking the average 15-minute coverage of the area of all Mondays in November as an example: (1) in the whole city (both inner city and suburban), the coverage rate of EMSs is 38.72% in morning rush hours, compared with 40% (±0.3%) in the remaining periods; (2) in the inner city, the coverage rate is 77.37% in morning rush hours, compared with 83% (±0.6%) in the remaining periods.*

9. L 283-288: The definition and selection of a precipitation event belongs to the methodology section.

Response: We agree with the reviewer's suggestion, and we moved this part to 2.2.2 in the revised manuscript. Please refer to Lines 155-161 in Pages 7-8.

10. 4.1.1. When I first read that you average out the total precipitation, across the grid cells, I was skeptical whether this conceals local effects, as one might assume that certain parts of the city could hence flood more, and cause over-proportional traffic delays.

Also, I do not see an analysis of total precipitation on traffic speed, which I could imagine has quite an impact (while it is certainly important how strongly it rains in a given hour, it surely also matters how long it rains for causing pluvial flooding). Please elaborate more on both those aspects.

Please also use figure 3 in justifying your assumptions / method, as indeed, it seems that without the (explained) outliers, there seems to be not much of an impact on how much it rains for causing

travel reductions? This may be an argument in favor of your decision; however, it is somewhat unintuitive why there is so little effect.

In general, please comment more on the relationship between precipitation and urban pluvial flooding, and limitations of looking only at precipitation data without any hydrological modelling associated with it, that would link precipitation with the actual amount of water on the streets.

**Response**: We thank for reviewer's comment. First, we averaged the total precipitation because we intended to evaluate the correlation of precipitation and overall traffic speed reduction in the city. And the spatial difference of precipitation was illustrated in Figure 4. We admit that could lead to conceal the local effects. This limitation was mainly due to the lack of high-resolution precipitation data. There are only 175 grids of precipitation data in Beijing, and we couldn't analyze the local effect on that scale. And we added more discussion on that in lines 487-492, page 25:

*There are also some limitations in this study. First, we averaged the traffic speed reduction rate of all the roads in the city, as well as the precipitation data, which could conceal congestion hotspots. In further studies, with higher resolution precipitation, along with corresponding traffic data, we could narrow the scale to blocks, pay more attention to local congestions, and analyze the correlation of precipitation and traffic speed on a finer scale.*

Second, we apologize for the ambiguity of the precipitation unit. Actually, the index of precipitation we used is the total precipitation during the selected time period, instead of the intensity of precipitation. Because the unit of precipitation we took was mm/2h, and in case that we only focus on the morning rush hours period (7:00-9:00), the index represents the total precipitation in these 2 hours. To avoid confusion, we mentioned it again in lines 287-288, page 13:

*The unit of precipitation data is mm/2h, which indicates the total precipitation in the 2 hours of morning rush hours.*

Besides, how long it rains was hard to be accurate because the time resolution of precipitation is 0.5 hours, and the morning rush hours period is only 2 hours in total.

Third, Figure 3 shows that there are 4 days that has more speed reduction, and the rest days seems relatively normal. We think the reason of this result is that the days with heavy rain or snow were rare. But when the small rain encountered the specific date, such as teacher's day, the impact would be amplified. The limitation is that we only have one-year traffic data, we believe with a longer time series data, the further results could be more significant.

Fourth, we agree that run a hydrological pluvial flood simulation could better link the precipitation, waterlogging and traffic congestion. However, run a flood simulation need a high-resolution DSM data of the city, which is very hard to get in China. We therefore link the precipitation and traffic directly with the analysis of ground-truth data. And we add some discussion about this in lines 500-503, page 25-26:

*Due to the lack of high-resolution DSM data, we didn't run a hydrological flood simulation in Beijing, which could reveal the relationship of precipitation and the actual amount of water on the streets. And this could be improved in the future studies with more high-resolution topographic data.*

11. Figure 4 is basically not commented and further analyzed. Please elaborate more, what one can see.

**Response**: We thank for reviewer's comment. We have elaborated the description of Figure 4 in the revised manuscript in Lines 296-306, Pages 14.

*Figure 4 illustrates the spatial difference of traffic speed reduction and distribution of precipitation on precipitation days. A large number of red roads (with traffic speed reduction over 10 km/h) can be observed in the 4 days mentioned above. By comparing the distribution of precipitation and traffic speed reduction on different dates in Figure 4, it can be found that the precipitation in the four days with the most severe speed reduction was moderate, and the precipitation distribution of the whole city was relatively uniform. Compared with other rain days, although the precipitation on July 5, August 9 and September 19 was larger and concentrated in the inner city, the traffic speed reduction of the whole city was not as serious as the four days mentioned above, which may be caused by the decrease of people's willingness to travel with the increase of rain.*

**Grammar / Style**

1. In-line citations are ill-formatted (brackets around them), e.g. L 83, L 90, etc. Please format correctly. Also, some citations are CAPITALIZED.

**Response**: We thank for reviewer's comment. We have checked and corrected the format of citations.

2. L31:  towns   new sentence : Furthermore, …

**Response**: We thank for reviewer's comment. We have corrected the gramma mistake.

3. L 106: Could you briefly explain the term "waterlogging" (e.g. the saturation of ground with water), as it may not be clear to every single reader what is meant by this phenomenon.

**Response**: As suggested by the reviewer, we have added the explanation in Lines 68-69, Page 4 waterlogging (the phenomenon of a stagnant water disaster in an urban area due to heavy rainfall or continuous precipitation).

4. L145: Rather: "hit" by a rainstorm? They do not malevolently "attack".

**Response**: We thank for reviewer's comment. We have changed the verb from "attack" to "hit". (Line 56, Page 3)

5. L 145-150: You already gave quite a few examples of hazardous events in the introduction; also, this example does not fit the section "Study area". Please consider to delete it.

**Response**: We thank for reviewer's comment. We delete it.

6. Section labels are wrong. For instance, after 2.1 and 2.2. on p7, we see another section 2.1 and 2.2. on p8

**Response**: We thank for reviewer's comment. We have corrected the section labels.

7. L261: The term "population medical accessibility index" is a little bit cumbersome to read and understand. Could you perhaps think of a simpler, more descriptive term?

**Response**: We thank for reviewer's comment. We have changed the term to "total transfer time" in Lines 259-261, Page 12:

*The total transfer time is introduced to quantify the cumulative transfer time for each population grid based on its population size, which is the number of potential users of EMSs.*

---

## Author Comment (AC2)

**Referee #2**

**Summary:** This paper presents a study that evaluates the spatial accessibility of emergency medical services during inclement weather, including rain and snow, and measures the impact of precipitation on traffic speeds. It compares the accessibility of emergency services during inclement weather to a baseline value calculated two weeks before the event and two weeks after the event. The results highlight four days when emergency medical service accessibility particularly decreased. The study also shows that snow has a particularly large impact on emergency service accessibility. The study has the potential to provide a scientific basis for discussions with transportation and urban planners to improve access to emergency medical services, particularly in rural areas or areas with unequal conditions.

**General comments:**

1. The study includes examples of natural hazards and the difficulty of reaching emergency services in a timely manner (L. 55- 63). Can you provide the references for these examples?

**Response**: We thank for reviewer's comment. We have added the references for the news.

*Beijing Evening. (2012). Beijing rainstorm, 120 calls increased by 1/3, trauma and car accident injury increased significantly. Available at: http://news.sohu.com/20120722/n348746024.shtml (accessed 30 August 2021).*
*Jimu News. (2021). Ambulance was blocked when the rainstorm hit the city in Qujing, Yunnan Province, firefighters transferred the injured boy in a canoe. Available at: https://new.qq.com/rain/a/20210624A0AFSB00 (accessed 30 August 2021).*
*Shaanxi News. (2019). Ambulance stalled on a rainstorm night, firefighters helped them get to hospital in time. Available at: http://news.cnwest.com/bwyc/a/2019/08/01/17913208.html (accessed 30 August 2021).*

2. The study presents several case studies that use different models (L. 70 -104). Could you please summarize the research gaps in this area?

**Response**: We thank for reviewer's comment. We recomposed the whole part of review in Lines 100-105, Page 5.

*Most of these studies assume that roads are impassable or traffic speed has a certain degree of reduction when the flooded water depth reaches a specific depth, and further evaluate the impact of rainstorm on EMSs accessibility. Due to insufficient recorded traffic data, relatively few studies have been performed to analyze the impact of road access capacity on EMSs accessibility according to actual traffic speed variation.*

3. The text gives a good description of the resolution of the data used. In line 157, please define "inclement" and "normal" weather in the datasets. Is a little rain already considered bad weather?

**Response**: We thank for reviewer's comment. In this study, we set a rule that if the precipitation of more than 10 grids in Beijing is greater than 1.5 mm, it is considered an inclement weather scenario. And yes, we include the small rain in the scope of inclement weather, because the overall rainfall intensity was not high in the year of 2019. To make it clear, we have defined it in Lines 155-161, Pages 7-8:

*According to the China Meteorological Administration, the moderate rain is defined as the rainfall is 5.0~14.9 mm within 12 hours. We chose intermediate value of interval and average it to each hour. In this study, we set a rule that if the precipitation of more than 10 grids (over 5% area of the city) in Beijing is greater than 1.5 mm in 2 hours, it is considered a precipitation event.*

4. Some sentences are very long sentences and compromise readability:

- o  L. 22 – 25 ("and" is used twice in short succession)

- o  L. 50 – 55

- o  L. 74 – 79

- o  L. 91 – 95

- o  L. 117- 123

- o  L. 141 – 145

- o  L. 366 – 373

**Response**: Thank you for pointing this out. We have rephrased the long sentences to several short ones.

5. L. 77: Please refer to the correct citation style and do not capitalize references: "Jones and Bentham…".

**Response**: We thank for reviewer's comment. We have corrected the citations.

6. L. 100: Could you please write out the abbreviation "PF-prone" when it is first mentioned?

**Response**: We thank for reviewer's comment. PF refers to pluvial flooding, and we have revised it.

7. L. 132: Instead of referencing the link in brackets, please refer to the correct citation style.

**Response**: We thank for reviewer's comment. We have corrected the citations.

8. L. 145: "Beijing was attacked by a rainstorm…". Could you please paraphrase this sentence?

**Response**: We thank for reviewer's comment. We have revised the sentence by changing the verb from "attack" to "hit". (Line 56, page 3)

9. L. 162: Section 2 has a wrong numbering of the subsections. Should this be 2.2.1. instead of 2.1, 2.2.2. instead of 2.2?

**Response**: We thank for reviewer's comment. We did make a mistake in the section labels, and it was corrected in revision.

10. L. 171: Instead of referencing the link in brackets, please use the correct citation style.

**Response**: We thank for reviewer's comment. We corrected the citations.

11. L. 183: "third-level grade-A hospitals." Could you please provide a brief explanation of hospital classifications that might help readers if they are not familiar with it?

**Response**: We thank for reviewer's comment. We added the explanation of "third-level grade-A hospitals." in Lines 169-172, Page 8:

*The hospital point data were extracted from the online map point of interest (POI) data of Beijing in 2019. After coordinate correction and deduplication, it contains a total of 630 general hospitals, 76 of which are third-level grade-A hospitals (the highest level in the evaluation system of hospitals in mainland China).*

12. L. 255: Could you please write out the abbreviation of "OD", when it is first mentioned?

**Response**: We thank for reviewer's comment. "OD" refers to origin-destination, and we have added it in the revision in Line 251, Page 12.

13. L. 465: Could you rephrase the phrase "we could guess"?

**Response**: We thank for reviewer's comment. We rephrased it in lines 497-499, page 25:

*Under such precipitation conditions, the EMSs accessibility has been affected to a certain extent, and it would be much more difficult to get timely EMSs under even more extreme inclement weather condition.*

- **Figures:**

1. Could you please specify which software tools you used to create the figures?

**Response**: We thank for reviewer's comment. Figure 1, Figure 4, Figure 6, Figure 8 and Figure 10 were created in ArcGIS 10.8 and composed in CoreDrawX7. Figure 2 was created in CoreDrawX7. Figure 3, Figure 5 Figure 7 and Figure 9 were created in Excel 2019. We add the explanation in the Method section in lines 185-186, Page 17:

*Both service area analysis and OD Cost Matrix analysis are GIS-based, and was done in ArcGIS 10.8.*

2. L. 305: Figure 4 is difficult to read. Is it possible to highlight some particular days with observations?

**Response**: We thank for reviewer's comment. We added black borders to highlight the 4 special days.

[Figure]

***Figure 4.*** *Variation in drive speed and distribution of precipitation on selected precipitation days*

*(the 4 subfigures with black borders shows the 4 most affected scenarios)*

- **Discussion:**

1. L. 435: In the discussion, it would be good to refer to the previously mentioned studies in the introduction and draw a link: How does this work build on the previously published literature body? Where do the results align, where do they differ?

**Response**: We thank for reviewer's comment. We have added discussions on this in Lines 474-480,

Pages 24-25:

*In previous literature, simulation methods were widely used on the research on EMSs accessibility or traffic capacity under inclement weather. The ground-truth traffic data that covers every road in the whole city for a whole year in a row, was hardly used in the previous studies of the impact of weather on traffic and accessibility. And our study could be a good empirical verification in this field of study. The reduction extent of EMSs accessibility was close to previous studies. And we also found that snowfall may have a greater impact, which is hard to find out using flood simulation methods.*

2 L. 467: As next steps, you mention that future studies should consider data on "extreme precipitation" events. Are there other data analyses that can be done with the available data?

**Response**: We thank for reviewer's comment. The lack of traffic data limits the sample size we could analyze. We are trying to obtain a longer time series traffic data, and its corresponding precipitation data. If we could get more samples with bigger precipitation magnitude, we could analyze more extreme scenarios. And we add some discussion about this in Lines 494-496, Page 25:

*If with longer time series precipitation and traffic data, we could analyze the impact of precipitation magnitude to the traffic and accessibility, instead of simply dividing the days in a binary manner into inclement and non-inclement weather days.*

**Specific comments:**

1. L. 24: Although it is mentioned in the Abstract, "inclement weather" is quite general. Later, in the introduction, the study refers to "rain or snow" (line 51). How much rain or snow is considered inclement weather, or is a little rain already inclement weather?

**Response**: We thank for reviewer's comment. We have mentioned it in Lines 155-161, Pages 7-8 in the revision:

*According to the China Meteorological Administration, the moderate rain is defined as the rainfall is 5.0~14.9 mm within 12 hours. We chose intermediate value of interval and average it to each hour. In this study, we set a rule that if the precipitation of more than 10 grids (over 5% area of the city) in Beijing is greater than 1.5 mm in 2 hours, it is considered a precipitation event.*

2. L. 63 - 65: Since this is a very general context, could you please provide some more references?

**Response**: We thank for reviewer's comment. Thank you for this comment. We have added the reference to this sentence:
*Huber, D. G., and Gulledge, J.: Extreme weather and climate change: understanding the link, managing the risk, Pew Center on Global Climate Change Arlington, 2011.*

*Stott, P.: How climate change affects extreme weather events, SCIENCE, 352, 1517-1518,2016.*
*Stott, P. A., Christidis, N., Otto, F. E., Sun, Y., Vanderlinden, J. P., van Oldenborgh, G. J., Vautard, R., von Storch, H., Walton, P., and Yiou, P.: Attribution of extreme weather and climate-related events, Wiley Interdisciplinary Reviews: Climate Change, 7, 23-41,2016.*

3. L. 78: Could you please name some references that use the 2SFCA method?

**Response**: We thank for reviewer's comment. We have added the reference to 2SFCA methods:

*Chen, X., and Jia, P.: A comparative analysis of accessibility measures by the two-step floating catchment area (2sfca) method, INT J GEOGR INF SCI, 33, 1739-1758,2019.*
*Kanuganti, S., Sarkar, A. K., and Singh, A. P.: Quantifying accessibility to health care using two-step floating catchment area method (2sfca): a case study in rajasthan, Transportation Research Procedia, 17, 391-399,2016.*
*Li, M., Kwan, M., Chen, J., Wang, J., Yin, J., and Yu, D.: Measuring emergency medical service (ems) accessibility with the effect of city dynamics in a 100-year pluvial flood scenario, CITIES, 117, http://doi.org/10.1016/j.cities.2021.103314, 2021b.*

4. L. 112 - 113: Could you please state the contribution of the study more clearly?

**Response**: We thank for reviewer's comment. We have elaborated the contribution of our study in Lines 110-113, Page 5 and in Lines 480-486 Page 25:

*Our study provides an approach for evaluating the effectiveness and fairness of EMSs based on ground-truth traffic data, and the results can not only provide reference for the optimization of EMSs in Beijing, but also provide reference cases for other cities, which has a great practical significance. The results from this study provide a scientific reference for city planning departments in Beijing to optimize the site selection of emergency service facilities and get prepared for traffic dispersion on inclement weather. The relevant methods mentioned in this paper can also be popularized and easily applied to other cities once traffic data or empirical formulas regarding the impact of inclement weather on road traffic can be obtained.*

5. L. 157: Can you give a brief description of the road network topology?

**Response**: We thank for reviewer's comment. We have added the description of the road network topology in Lines 135-139, Page 7:

*Based on traffic data and meteorological data, we could build a topology road network (using node and edge primitives to describe interconnected linear features (roads) and points (roads junctions) on a map) with transfer time as impendence under inclement weather conditions and corresponding normal weather conditions.*

6. L. 203: How many days with precipitation were included in the sample?

**Response**:We thank for reviewer's comment. There are 19 working days of rainfall and 3 working days of snowfall were selected. To make it clear, we have added the following description in Lines 159-160, Page 8:

*The average precipitation of the whole city on each date is averaged by the precipitation of all grids. In 2019, 19 working days of rainfall and 3 working days of snowfall were selected.*

7. L. 298: The analysis focuses on specific holidays (July 1$^{st}$, September 10$^{th}$). How transferable are the results of your study to other days?

**Response**:We thank for reviewer's comment. Our study provides a general method to evaluate the EMSs accessibility, it's suitable in both holidays and workdays. These specific days like July 1st or September 10th may affect the traffic flow, which has an amplification effect on the traffic congestion caused by inclement weather. And there could be more days that would change the normal traffic flow, and when they encounter the inclement weather, there are potential risks of decrease of traffic efficiency and EMSs accessibility, which should be given sufficient attention. We have added this in discuss section in lines 467-470, page 24:

*Some specific days may affect the traffic flow, which has an amplification effect on the traffic congestion caused by inclement weather. When they encounter the inclement weather, there are potential risks of decrease of traffic efficiency and EMSs accessibility, which should be given sufficient attention.*

8. L. 254: "population medical accessibility index". The term can be a little difficult to understand. Can you briefly explain the term in more detail?

**Response**:We thank for reviewer's comment. We changed it into "total transfer time", which describe the total time would need if every person in this grid did once transfer process.

**Technical corrections:**

- L. 31: "towns with lower baseline EMS accessibility **are** more vulnerable to inclement weather. Furthermore,".

- L. 53: For quotations in continuous text, please insert a space in between the text and the reference: "The efficiency of emergency services is highly vulnerable to inclement weather conditions[...], and sometimes block roads completel**y (A**garwal et al., 2006;…"

- L. 152: For quotations in continuous text, please insert a space in between: "Andersson and Stålhult (2014) used network analysis"

- L. 188: How about phrasing the sentence: "The data records present the population size" or "The data records depict the population size…"?

- L. 192: How about phrasing the sentence: "Figure 2 presents" or "Figure 2 illustrates"?

- L. 315: Is it "In which the 15-min EMS coverage rate **reduced** by …"?

- L. 319: "…which **led** to a significant reduction in overall EMS coverage…"

- L. 418: Here, should it be "within the **Sixth** Ring Road extent"? Later, in line 363 and in line 365, the text refers to "within the Sixth Ring road".

- L. 428: "…were almost no regions where the population medical accessibility index **decreased**."

**Response**:  We thank for reviewer's comment. They all have been revised.

---

## Author Comment (AC3)

**Referee #3**

This empirical study investigates the impact of inclement weather on the time emergency medical response (EMS) time intervals for the city of Beijing. It is broken into two stages. Firstly, to explore the impact of inclement weather (i.e., precipitation) on traffic and EMS accessibility to come with the worst-case scenarios of the year 2029 (i.e., days including different times per day). Secondly, to evaluate EMS accessibility under the identified worst-case scenario and evaluate the distribution of EMS with particular focus on the difference in population and road network distribution between urban and suburban areas. The study can be useful to identify the scenarios needing improvement to ensure more fair access to EMSs for populated cities. The paper is generally well-written and easy to read but can be improve in terms of clarity.

1. The abstract. It seems to overlook a key impact that seems important from the results: The day of snowfall seems to have more significant impact that the days of rainfall (among the worst-case scenario considered). Can the authors add a mention about this fact somewhere in the abstract.

**Response**: We thank for reviewer's comment. We added one sentence in abstract in Lines 31, Page 2:

*And snowfall has a greater impact on the accessibility of EMS than rainfall.*

2. The introduction. In Line 107, Please be specific on what is meant by "The latter". It can be more effectively used to also introduce to lay reader some common terms that will be occurring later, such as "coverage area" and "waterlogging".

**Response**: We thank for reviewer's comment. We revised the whole paragraph in Lines 73-105, Page 4-5:

*The spatial accessibility of EMSs is defined by the travel impedance (distance or time) between service locations and the scene (Guagliardo, 2004). A large body of research on spatial accessibility is concerned with access to hospitals (Luo and Wang, 2003; Mao and Nekorchuk, 2013; Pan et al., 2018; Yang et al., 2020; Yin et al., 2021) and first-aid stations (Hashtarkhani et al., 2020; Jones and Bentham, 1995; Shin and Lee, 2018). To measure the EMSs accessibility, the two-step floating catchment area (2SFCA) method is one of the common methods (Chen and Jia, 2019; Kanuganti et al., 2016; Li et al., 2021; Luo and Qi, 2009). The 2SFCA method considers accessibility to be mediated by not only the distance decay but also the interactions between supply and demand (Chen and Jia, 2019), which is more suitable for normal scenarios. While for studies focusing on the influence of inclement weather on EMSs, people concern more about the transportation situation, instead of the interaction between supply and demand. The coverage analysis method (Coles et al., 2017; Green et al., 2017; Yu et al., 2020) or shortest path analysis method(Albano et al., 2014; Andersson and Stålhult, 2014) are more widely used. These methods could better characterize the reduction of accessibility caused by the road service degradation. For example, Yu et al. (2020) analyzed the accessibility of emergency service in England and identified vulnerability hotspots by*

*quantifying the EMSs coverage of area and population within different time radii under different flood scenarios; Coles et al. (2017) measured the travel time and service area coverage of EMSs in York, UK under flood scenarios by using FloodMap-HydroInundation2D to model flood inundation; Yin et al. (2021) assessed the vulnerability of EMSs to surface water flooding in Shanghai, China by quantifying accessibility in terms of service area, response time, and population coverage, considering four temporal scenarios (nighttime, evening, daytime, and morning–evening peak) of average drive speeds based on a real-time traffic analysis from GPS big data; Andersson and Stålhult (2014) used network analysis methods to generate the shortest paths from hospitals to various administrative areas in Manila, Philippines, and evaluated the impact of different flood events on these paths. Most of these studies assumed that roads are impassable or traffic speed has a certain degree of reduction when the flooded water depth reaches a specific depth, and further evaluated the impact of rainstorm on EMSs accessibility. Due to insufficient recorded traffic data, relatively few studies have been performed to analyze the impact of road access capacity on EMSs accessibility according to actual traffic speed variation.*

3. What platform was used to "Combining the topology road network with medical facility locations and the distribution of the population, we could further analyze the spatial accessibility to EMSs." Was this work GIS based? What was the tool employed?

**Response**: We thank for reviewer's comment. In this study, we used ArcGIS 10.8 to run the analysis. We added that in the method section in Lines 185-186, page 9:
*Both service area analysis and OD Cost Matrix analysis are GIS-based, and was done in ArcGIS 10.8.*

3. Line 162: the sub-section numbering here down to line 189 is missing a third digit. I suppose it should be 2.2.1, 2.2.2, 2.2.3 and 2.2.4.

**Response**: We thank for reviewer's comment. We apologize for the mistake in the section labels, and it was corrected in revision.

Methodology.

1. As in any study, there should some assumption made by the authors and aspects that were not addressed. A mention of these would be useful.

**Response**: We thank for reviewer's comment. In this study, we have these assumptions in lines 187-195, page 9:

1. *The ambulances move at the average speed all the time and would always take the shortest path in space.*
2. *In network analysis, the location of facilities is approximately considered to be on the nearest road point vertically.*
3. *In OD analysis, we use the centroid as the origin point to represent the whole grid, and the*

*shortest path to hospital is the same within the grid.*

4. *The prehospital EMSs is divided into two parts: the ambulances depart from the first-aid station to the scene and from the scene to the nearest hospital. The case where patients transfer directly from the scene to an EMS facility via private transportation will not be considered in this study.*

2. Line 232: Is there any citation that you can use to justify the choice of the 15-minute arrival time?

**Response**: We thank for reviewer's comment. In previous study, considering various response time targets, three service zones lying within 8-, 12-, and 15-minute travel are specified for each individual EMS station. And the coverage areas all decreased under the impact of flood. So, we chose 15-minute arrival time. In the revised manuscript, we added the reference:

*Yin, J., Yu, D., and Liao, B.: A city-scale assessment of emergency response accessibility to vulnerable populations and facilities under normal and pluvial flood conditions for shanghai, china, ENVIRON PLAN B-URBAN, 48, 2239-2253, http://doi.org/10.1177/2399808320971304, 2021*

5. Line 255: please define OD. Does it stand for Origin-Destination?

**Response**: We thank for reviewer's comment. Yes, "OD" refers to origin-destination, and we added it in Lines 250-251, Page12:

*The shortest transfer time is calculated by the OD (Origin-destination) cost matrix analysis method.*

6. Line 258-259: More discussions on the calculation cost is welcome so one can justify the rationale behind increasing the resolution from 100 and 1000 m. How much would this impact the predictions vs. reduce the cost for the analysis?

**Response**: We thank for reviewer's comment. The choice of roads to the nearest hospital won't be greatly impact because the distance to the nearest to the nearest hospital is normally more than 1km in most areas, so increasing the resolution would not affect the overall pattern of spatial difference of accessibility, but would the cost of calculation would reduce to 1/100, because every grid needs to calculate the shortest route and transfer time to every hospital. After the aggregation of population grids, there are about 25,000 origin points, and we need to calculate the shortest travel path between every origin point and hospital and select a closest hospital for each origin point. The calculation can be done in about 10~15 minutes for each scenario. If we don't do the aggregation, the number origin points would be about 2,500,000, so the calculation would be more than 1000 minutes.

7. Line 265: there is always a mention of a grid. Should this be meaning that the analysis was GIS based?

**Response**: We thank for reviewer's comment. The analysis is GIS based. We have added this information in Lines 184-185, Page 9:

*Both service area analysis and OD Cost Matrix analysis are GIS-based, and was done in ArcGIS*

*10.8.*

6. Line 306: Figure 4 deserves a discussion as such in 4.1.1 before moving to 4.1.2 and quoting it there. I guess it was used to support further the choice for the days considered as worst-case scenario.

**Response**: We thank for reviewer's comment. We add some elaboration in lines 296-299, page 14:

*Figure 4 illustrates the spatial difference of traffic speed reduction and distribution of precipitation on precipitation days. A large number of red roads (with traffic speed reduction over 10 km/h) can be observed in the 4 days mentioned above. By comparing the distribution of precipitation and traffic speed reduction on different dates in Figure 4, it can be found that the precipitation in the four days with the most severe speed reduction was moderate, and the precipitation distribution of the whole city was relatively uniform. Compared with other rain days, although the precipitation on July 5, August 9 and September 19 was larger and concentrated in the inner city, the traffic speed reduction of the whole city was not as serious as the four days mentioned above, which may be caused by the decrease of people's willingness to travel with the increase of rain.*

7. Line 331: "The results demonstrate …" what results? Any figures I should be looking at? Or, from which equation? Are you talking about the "coverage rates". Please specify.

**Response**: We thank for reviewer's comment. The results refer to the spatial difference of population coverage. We added a new figure to illustrate it better. The new figure has 2 rows, using the first row minus the second row is the variation that Figure 6 (the order number becomes Figure 7 now) shows, as can be found in Page 17:

[Figure]

**Figure 6.** The EMSs coverage rate of population in townships under the inclement weather condition

and normal weather condition on 1st July, 9th July, 10th September and 16th December

8. Line 359: The clarity of the sub-figures in Figure 7 can be significantly improved. Same for Figure 8.

**Response**: We thank for reviewer's comment. We have enlarged the font in the figures to make it clearer. The figure number changed in the revision of manuscript.

[Figure]

(a)  (b)

(c)  (d)

**Figure 8.** The correlation between the baseline EMS coverage rate of population and its reduction

percentage in inclement weather. (a) 1st July, (b) 9th July, (c) 10th September, and (d) 16th December

(a)  (b)

[Figure]

(c)                                    (d)

**Figure 10.** The correlation between the baseline transfer time to hospitals and the increased transfer

time in inclement weather. (a) 1ˢᵗ July, (b) 9ᵗʰ July, (c) 10ᵗʰ September, and (d) 16ᵗʰ December

9. Overall, the snowfall seems to have the greatest impact and it useful to highlight in key locations including the abstract and conclusions. This could be hinting at the fact that such a study is more of relevance to cities affected by snowfall.

**Response**: We thank for reviewer's comment. We have added that in abstract and conclusions in Lines 31-32, page 2 and Lines 460-461, page 24:

*And snowfall has a greater impact on the accessibility of EMSs than rainfall.*
*Besides, snowfall has a greater impact on EMSs accessibility than rainfall.*

---

## Author Response (AR2)

**Spatial accessibility of emergency medical services under inclement weather: A case study in Beijing, China**

**Ref: nhess-2022-218**

We would like to thank the editor and reviewers for the thorough reading of the manuscript and the valuable remarks that helped us to improve the manuscript. We have revised the manuscript carefully according to the reviewer's comments, and have incorporated the suggestions into the revised manuscript.

The notes below provide a point-by-point response to each comment from the referees. The texts with blue font are the reviewer's original comments, the texts with black font are authors' responses. We have incorporated most of the suggestions made by the reviewers. Those changes are highlighted within the manuscript. If there is any question addressed unclearly or unsatisfied, we are always willing to make a second revision based on reviewer's comments. Thank you again for the opportunity to be considered for publication in *Natural Hazards and Earth System Sciences*.

**Referee #1**

The authors address most of my comments and I am thankful to this. However, I feel acknowledging some lesson learnt about the applicability of the proposed methodology to a different city, and the limitation of the seemly overoptimistic assumption of "15-minutes arrival time" are necessary to have this contribution as realistic as possible. Details can be found below with reference to my "Original comment" followed by the "Authors' response" and then by the "New reviewer's comments" in relation to the "Authors' response".

Original comment: "2. Line 232: Is there any citation that you can use to justify the choice of the 15-minute arrival time?"

Authors' response: "[…] In previous study, considering various response time targets, three service zones lying within 8-, 12-, and 15-minute travel are specified for each individual EMS station. And the coverage areas all decreased under the impact of flood.

So, we chose 15-minute arrival time. In the revised manuscript, we added the reference:

Yin, J., Yu, D., and Liao, B.: A city-scale assessment of emergency response accessibility to vulnerable populations and facilities under normal and pluvial flood conditions for shanghai, china, ENVIRON PLAN B-URBAN, 48, 2239-2253, http://doi.org/10.1177/2399808320971304, 2021"

New reviewer's comments: In the "previous study" of Yin et al. (2021) – cited above – the investigators seem to have done a similar study but for Shanghai. On this, I have two comments:

(1) the author should discuss the findings of Yin et al. (2021) first in the introduction of this paper. Is the same method applied from a different city? If so, a reflection on what do learn from it as compared to the other city, Shanghai, would be useful – in the conclusions; and

**Response**: We thank for reviewer's comment. The method of quantifying the EMS accessibility is similar in our study and the paper of Yin et al. (2021): we both used the service area analysis based on ArcGIS Network Analysis. The main differences between our study and theirs is that we used the real recorded transportation data to analyze the influence of inclement weather on EMS accessibility, which is actually one of the important novelties of our study. In Yin's research, they used flood simulation model to build flood scenarios and set 4 drive speed limit level to build different transportation scenarios. We have added the findings of Yin et al. (2021) in the introduction part and further emphasize the research differences in Lines 94-98, Page 5:

*Yin et al. (2021) assessed the vulnerability of EMSs to surface water flooding in Shanghai, China by quantifying accessibility in terms of service area, population coverage and response time, and the results show that EMS coverage could decrease up to 13% under 100-year surface water flooding;*

In Yin's study, they found that "compared with normal operating condition, 5- and 20-year pluvial flooding both exerted very minor and even negligible impacts on the change of service area (less than 1%). The impact of 100-year surface water flooding is more pronounced (up to 13%)." Compared to our study, because the inclement weather days that we analyzed in 2019 didn't have quite high precipitation, so in our results, the most of the coverage area reductions were only around 0%-3%, which is consistent with their conclusions to some extent.

We also added some comparisons to the results in Lines 330-336, Page 16:

*Consistent with the pattern of the traffic speed reduction, the worst loss of coverage rate also occurred on three rainy days: 1st July (Mon), 9th July (Tue), and 10th September (Tue), and one snowy day: 16th December (Mon), in which the 15-minute EMS coverage rate reduced by 4.6%, 5.6%, 4.2% and 13.3%. The rest days didn't have obvious coverage reduction. Combined with the spatial distribution of precipitation and traffic variation (Figure 4), the snowfall on December 16th caused a large traffic speed reduction of the suburban roads, which led to a significant reduction in overall EMS coverage. In previous studies, Yin et al. (2021) found that 5- and 20-year pluvial flooding both exerted less than 1% reduction in EMSs coverage rate of Shanghai, China; Coles et al. (2017) found that the coverage of Fire and*

*Rescue Stations services showed a 6% reduction overall under their modelled floods events in York, UK. In our study, the precipitation was less than 3mm/2h, and the corresponding coverage reduction was less than 3%, except for the special four days. The results are comparable to previous findings. In the following,* we chose these four days as the worst weather scenario of the year and analyzed the spatial differences of medical accessibility in the whole city.

(2) the paper of Yin et al. (2021), which involves one of the co-authors, does not really justify the choice of the 15-minutes arrival time. Having looked at the NHS website for the UK, it seems like 8 to 19 minutes is an expected time window for an ambulance to arrive in normal weather conditions. Therefore, should the authors keep the choice of the "the 15-minute arrival time", this has to be discussed as part of a limitation sub-section or as part of the "assumptions" made. In fact, the 15-minutes travel time for an ambulance under flooding or snowfall conditions may be overly optimistic. If so, this should be acknowledged.

**Response**: We thank for reviewer's comment. In China, there is no national legislative requirements for emergency response time. However, in Beijing, according to the data from the report of Beijing Municipal Health Commission, the average response time of pre-hospital emergency treatment is about 15 minutes in 2022. Therefore, we chose 15-min as the boundary in our study. We have added the news report to the references.

*Beijing Youth Daily. Available at: https://t.ynet.cn/baijia/33458913.html (lase access: 11 February 2023),2022.*

Indeed, the circumstance would be different in other cities like London, UK. We admit that we were overoptimistic to the applicability of the proposed methodology to a different city. Therefore, we added the following explanation to the discussion in Lines 504-508, Pages 25-26, and the assumption in Lines 195-199, Page 9:

*Fourth, we used the "15-minutes arrival time" as a main boundary in this study, however, the proper response time would vary in different countries or cities. So, the setting of response time boundary should be adjusted considering the actual situation of the city when the method in this paper is applied to other cities.*

*In this study, we made the following assumptions: ...... (5) According to the report by Beijing Municipal Health Commission, the average response time of pre-hospital emergency treatment in Beijing is about 15 minutes for the year of 2022. We therefore chose 15-min as the boundary of EMSs response time in our study. (Beijing Youth, 2022).*

**Referee #2**

The authors implemented valuable amendments and clarifications, which greatly improved the manuscript. Limitations of the study are also explained better and put into context.
Altogether, the study provides an interesting first proof of concept on how to integrate various forms of empirical data into the research question of EMS accessibility under adverse weather conditions, and conveniently summarises a range of concepts, that can be more refined in the future.

A few technical (and grammar-related) comments remain, together with some minor content-related questions:

Grammar and style: The following points are only anecdotally collected instances where amendments should be made, please consider proof-reading by a native speaker for a smoother reading flow.

L28: the word "citywide" does not fit here.
L 61: 1.5 to 2 hours; wouldn't --> would not (please check the manuscript for similar informally-sounding instances)
L62: on usual --> usually
L82 - 85: people are more concerned about (... ) than about.
L155: the moderate rain is defined as the rainfall is 5.0~14.9 mm within 12 hours. --> moderate rain (without "the") is defined as rainfall between 5.0 and 14.9 mm within 12 hours (?)

**Response**: We thank for reviewer's comment. They all have been revised in the revised manuscript.

Lines 28-29: *the area in the city that could get EMSs within 15 minutes would decrease by 13% compared to normal scenario.*
Lines 62-63: *Usually, the transfer time would not be more than 1 hour.*
Lines 84-86: *people are more concerned about the transportation situation, instead of the interaction between supply and demand.*
Lines 156-157: *moderate rain is defined as the rainfall is 5.0~14.9 mm per 12 hours.*

To Section 3.3:
True, interpreting the centroid of the aggregated 1x1km population grid as a "sample" seems a valid interpretation - especially for such a large city as Beijing, where this still results in >16k population nodes on the graph.
As a thought for future studies, though: looking at the topology of the studied network, health facilities, roads and population density in the inner city are much denser than in the suburbs. Hence a varying resolution could have been applied with a finer grid in the

heavily populated center, and a more coarse grid towards the outskirts. This could still keep the number of OD nodes equally reduced, while capturing more of the dynamics in a metropolis with varying population and infrastructure densities.

**Response**: We thank for reviewer's comment. A varying resolution, with a finer grid in the heavily populated center and a coarser grid towards the outskirts, would help to capture more of the dynamics in a metropolis. We are willing to try this method in future research. We added this in the discussion part in Lines 508-511, Page 26:

*Fifth, we aggregated the population grid evenly in the city. If a varying resolution could have been applied with a finer grid in the heavily populated center, and a coarser grid towards the outskirts, it may capture more of the dynamics in a metropolis with varying population and infrastructure densities.*

Results 4.2.1 - Figure 8
In lines 372-374 you write that "the results reveal that the population of the towns with low baseline EMS coverage rate would lose more EMS coverage under inclement weather, especially on snowy day." While the one hypothesis which you discuss to explain this, namely the lower redundancy of possible access roads in rural areas, makes sense, there is a second point to this: the plots in figure 8 show relative (!) changes in EMS coverage rates. Hence, if base accessibility was already low, it is only natural that relative changes for low numbers are very big (for instance, if in a suburban zone, base coverage rate of EMS was around 20%, and it dropped to 15% during inclement weather, this would show up as a 25% relative reduction in your plot), whereas if in an inner-city zone coverage drops by the same percentage (say, from 100% to 95%), this would show up in your plot only as a 5% decrease. This has hence nothing to do with road topology and physical accessibility, but is an artefact of the display method.

I would suggest having a look at a modified version of the plots in figure 8, plotting base rate EMS coverage against daily EMS coverage rate (i.e. in absolute % numbers, not in relative reduction), to see if this over-proportional affectedness persists in such a plot.

**Response**: We thank for reviewer's comment. We fully understand the concerns of the reviewer.

The absolute reduction and the relative reduction are two different aspects reflecting the inequity of town's EMSs accessibility. As the reviewer pointed out, we first evaluated the absolute change in our study. In Figure 7, we used the spatial distribution map to present the absolute reduction of each town. We can see that lots of areas in suburban did experience severe decrease in population coverage under inclement weather conditions.

Meanwhile, we also pay attention to the relative reduction, which is also a very important indicator. Comparing to the absolute reduction, this could better reflect town's changes relative to themselves. For example, if in a suburban town, base coverage rate was about 10% and it dropped to 0% during inclement weather, this town would be almost completely unavailable to EMSs. And these towns should be the key areas in the planning of infrastructure construction. However, if in an urban town, the coverage rate dropped from 100% to 90% during inclement weather, it still can obtain relatively high of EMSs coverage. So, we still keep the display of the relative reduction of the EMSs coverage.

Lastly, both figures 8 and 10 are not extremely insightful, but take quite a bit of space. I would consider either merging the multi-panels, or moving them to the appendix.

**Response**: We thank for reviewer's comment. We have moved both figure 8 and 10 to the appendix.

**Referee #3**

Specific comments:
L. 27/ 80: The definition of "normal": Can you provide an explanation to the reader to facilitate their understanding?

• L. 27: "Under inclement weather scenario, the area in the citywide that could get EMSs within 15 minutes would decrease by 13% compared to normal scenario, while in some suburban townships, the population that could get 15-min EMSs would decrease by 40%."
• L. 80: "The 2SFCAmethod considers accessibility to be mediated by not only the distance decay but also the interactions between supply and demand (Chen and Jia, 2019), which is more suitable for normal scenarios"

**Response**: We thank for reviewer's comment. "Normal" refers to "the average state of weekdays without precipitation" in this paper. We have added the explanation where the word first appeared in Lines 28-30, Page 2:

*the area in the city that could get EMSs within 15 minutes would decrease by 13% compared to normal scenario (the average state of weekdays without precipitation)*

Further explanation required:
• L. 51: "Because inclement weather conditions would reduce road capacity, increase transfer time, and sometimes block roads completely (Agarwal et al., 2006; Chang et al., 2013; Cools et al., 2010; Suarez et al., 2005; Zhang and Chen, 2019), resulting in

reduced spatial accessibility and delayed the response time of EMSs." --> This sentence seems to be somewhat incomplete. In the previous version, this was part of the previous sentence.

**Response**: We thank for reviewer's comment. The sentence in the previous version was split into two sentences because the sentence was considered too long to be readable. We tried to revise it again in Line 49-55, Page 3:

*The efficiency of emergency services is highly vulnerable to inclement weather conditions such as rain, snow, frog, etc. The reason why inclement weather conditions would reduce the efficiency of emergency services is that inclement weather conditions would reduce road capacity, increase transfer time, and sometimes block roads completely, which leads to the reduction of spatial accessibility and delay of response time.*

• L. 82: "While for studies focusing on the influence of inclement weather on EMSs, people concern more about the transportation situation, instead of the interaction between supply and demand." --> What is meant by "for"?

**Response**: We thank for reviewer's comment. We have revised the sentences in Lines 83-85, Page 4:

*While in the studies focusing on the influence of inclement weather on EMSs, people are more concerned about the transportation situation*

• L. 321: "Combined with the spatial distribution of precipitation and traffic variation (Figure 4) to analyse, …" --> "to analyse" does not fit in the sentence. Can you leave that out?

**Response**: We thank for reviewer's comment. We have removed "to analyze" in this sentence in Lines 327-330, Page16.

*Combined with the spatial distribution of precipitation and traffic variation (Figure 4), the snowfall on December 16th caused a large traffic speed reduction of the suburban roads, which led to a significant reduction in overall EMS coverage.*

• L. 324: "Therefore, we chose these four days as the worst weather scenario of the year and analysis the…" --> Do you mean "and analysed the…"?

**Response**: We thank for reviewer's comment. Yes, it should be "analyzed" and we corrected the mistake in Lines 336-338, Page16.

*In the following, we chose these four days as the worst weather scenario of the year and analyzed the spatial differences of medical accessibility in the whole city.*

• L. 151: "The meteorological data utilized in this paper are TRMM precipitation" --> Could you write out the full term in case readers are not familiar with the abbreviation?

**Response**: We thank for reviewer's comment. TRMM refers to Tropical Rainfall Measuring Mission. We added this in Lines 152-153, Page 7:

*The meteorological data utilized in this paper are TRMM (Tropical Rainfall Measuring Mission) precipitation data obtained from NASA.*

• L. 263: "population medical accessibility index..." --> You have previously introduced the "total transfer time", but use "population medical accessibility index" here.

**Response**: We thank for reviewer's comment. The mistake has been revised in Lines 266-267, Page 13:

*For each population grid centroid i, its total transfer time (T) is calculated by eq.(4)*

• L. 500: "Third, due to the lack of high-resolution DSM data…" --? Could you write out the full term in case readers are not familiar with the abbreviation?

**Response**: We thank for reviewer's comment. DSM refers to Digital Surface Model. We added this in Lines 500-502, Page 25:

*due to the lack of high-resolution DSM (Digital Surface Model) data, we didn't run a hydrological flood simulation in Beijing*

• L. 478: "The reduction extent of EMSs accessibility was close to previous studies." --> Can you briefly mention the other studies?

**Response**: We thank for reviewer's comment. We have added the reference for this sentence in Lines 478-479, Page 24, and we have added brief comparison in our results in Lines 330-334, Pages 16:

*The reduction extent of EMSs accessibility was close to previous studies (Yin et al., 2021; Coles et al., 2017).*

*In previous studies, Yin et al. (2021) found that 5- and 20-year pluvial flooding both exerted less than 1% reduction in EMSs coverage rate of Shanghai, China; Coles et al. (2017) found that the coverage of Fire and Rescue Stations services showed a 6% reduction overall under their modelled floods events in York, UK.*

Figures:
• Could you indicate the software used to produce each figure, such as figures 3, 5, 8, 10?
• Could you also indicate the name of the GIS software used to produce the figures 1, 4, 6, 7, 9, and 11? Can this information be included in the caption and/or appendix?
**Response**: We thank for reviewer's comment. Figures 3, 5, 8, 10 were made in Excel 2016. Figures 1, 4, 6, 7, 9, and 11 were made in ArcGIS 10.8. And we have added the descriptions after each figure title.

Technical corrections:
• L. 28: "… the area in the citywide that could get EMs" --> Do you mean "the area in the city" or "the area in the citywide network"?
• L. 33: "Furthermore, towns with lower baseline EMSs accessibility is more vulnerable to inclement weather." --> "towns… are".
• L. 185: "Both service area analysis and OD Cost Matrix analysis are GIS-based, and was done in ArcGIS 10.8" --> "Both the service area analysis and the OD cost matrix analysis are GIS-based and were done in ArcGIS 10.8."?
• L. 269: "the diurnal variation in traffic can be divided into fourperiods" --> "four periods"?
• L. 483: "The relevant methods mentioned in this paper is also suitable for both holidays and workdays" --> "are suitable"?
• L. 494: "If with longer time series precipitation and traffic data, we could analyze the impact of precipitation magnitude to the traffic and accessibility,…" --> The sentence is somewhat cumbersome. Do you mean "If we had longer time series…"?
• L. 501: "we didn't run a hydrological" --> Could you write "did not"?
• L. 61: "1.5~2 hours for each evacuation during the rainstorm, while the transfer time wouldn't" --> Could you write "would not" ?
**Response**: We thank for reviewer's comment. We have corrected these technical mistakes based on reviewers' comments.

There are inconsistencies in line quotes. Sometimes a space is inserted between the end of the sentence and the reference, sometimes not. Could you please be consistent. Examples include:
• L. 46: "survival(Blackwell and Kaufman, 2002)."
• L. 68: "cities more frequently(Huber and Gulledge, 2011; ….)"
• L. 69: "problem of urban rainstorms and waterlogging(the…)"

• L. 86: "shortest path analysis method(Albano…"
• L. 201: "holiday effects(Cools et al., 2007), season…."
• L. 227: "area analysis of the 15-minute(Yin et al.,..."

**Response**: We thank for reviewer's comment. We have inserted spaces between the end of the sentence and the reference throughout the paper.

There are inconsistencies in the spacing between numbers and units. Could you please be consistent. Examples include:
• L. 65: "190.6mm", but line 58 mentions: "170 mm"

**Response**: We thank for reviewer's comment. We have unified the spacing between numbers and nits throughout the paper. For example, the "190.6mm" has been changed into "190.6 mm", and the number "170 mm" has been changed into "170.0 mm".

There are inconsistent spellings of the term hotspot. Examples include:
• L. 358: "the hot spots…", but lines 24/ 90 mention: "hotspots"

**Response**: We thank for reviewer's comment. We have corrected the inconsistencies in the revised manuscript.

There are inconsistencies in the use of words or numerals for numbers:
• L. 206: "…we would continue to look forward or backward until 4 baseline days are found. The average speed data of the four baseline days…" --> Use "four" consistently.
• L. 203: "For a given precipitation day, we search for the same day of week in the 2 weeks forward and backward to obtain the corresponding baseline days without precipitation." --> Use "two"

**Response**: We thank for reviewer's comment. We have corrected the inconsistencies in the revised manuscript.

Repetitions and very long sentence that affect readability.
• L. 57: "For example, on July 21, 2012, Beijing was hit by a rainstorm, with the average cumulative rainfall reaching 170 mm, caused 63 roads to be seriously flooded, and led to a one-third increase in the number of calls to the emergency center, and the transfer time of ambulances was significantly prolonged, taking approximately 1.5~2 hours for each evacuation during the rainstorm, while the transfer time wouldn't be more than 1 hour on usual (Wang et al., 2013; Beijing Evening,2012)."
• L. 359: "Compared with other districts in inner suburbs, such as Shunyi,Daxing, and Tongzhou, these areas are farther away from the city center and have less distribution of medical facilities and sparser road networks and more vulnerable to inclement weather, and these areas are also regions with a relatively higher proportion of the elderly population over the age of 80 in the total population." --> Also, the word "and" is used too often, which affects readability.

**Response**: We thank for reviewer's comment. We revised the sentences in Lines 58-64, Page3 and Lines 372-375, Pages 18-19:

*For example, on July 21, 2012, Beijing was hit by a rainstorm, with the average cumulative rainfall reaching 170.0 mm, caused 63 roads to be seriously flooded. This rainfall event led to a one-third increase in the number of calls to the emergency center, and the transfer time of ambulances was significantly prolonged, taking approximately 1.5~2 hours for each evacuation during the rainstorm. Usually, the transfer time would not be more than 1 hour. (Wang et al., 2013; Beijing Evening,2012)*

*The suburb areas, such as Shunyi, Daxing, and Tongzhou, are more vulnerable to inclement weather as they have less distribution of medical facilities and sparser road networks, as well have a relatively higher proportion of the elderly population over the age of 80.*

Could you paraphrase sentences that currently begin with "And"?
• L. 30: "And we found that snowfall has a greater impact on the accessibility of EMSs than rainfall."
• L. 383: "And on snowy days"
• L. 503: "And this could be improved"

**Response**: We thank for reviewer's comment. We removed the meaningless "and" in these sentences.

Citation style:
• L. 118: "According to the seventh national census (http://www.stats.gov.cn/tjsj/tjgb/rkpcgb/)" --> Could you please adhere to the citation style and do not include the web link in-line?
• L. 155: "According to the China Meteorological Administration" --> Could you please provide a reference for this source?
• L. 165: "The locations of these first-aid stations were obtained from the distribution map of first-aid stations published on the official website of the Beijing Emergency Center," --> Could you please provide a reference for this source?
• L. 168: "The hospital point data were extracted from the online map point of interest (POI) data of Beijing in 2019." --> Could you please provide a reference for this source?
• L. 175: "The demographic data of 2019 were obtained from WorldPop..." --> Could you please provide a reference for this source?

**Response**: We thank for reviewer's comment. We have added references in the revised manuscript as given below.

*National Bureau of Statistics. Bulletin of the National Population Census. Available at: http://www.stats.gov.cn/tjsj/tjgb/rkpcgb/ (last access: 11 February 2023), 2021.*

*China Meteorological Administration. Classification of precipitation. Available at: https://www.cma.gov.cn/2011xzt/2012zhuant/20120928_1_1_1_1/2010052703/201212/t20121212_195616.html (last access: 11 February 2023), 2012.*

*Beijing Emergency Medical Center. Available at: https://beijing120.com/channel/184 (last access: 30 August 2021).*

*Gaode Maps. Available at: https://lbs.amap.com/api/webservice/guide/api/search (lase access: 30 August 2021).*

*WorldPop (www.worldpop.org - School of Geography and Environmental Science, University of Southampton; Department of Geography and Geosciences, University of Louisville; Departement de Geographie, Universite de Namur) and Center for International Earth Science Information Network (CIESIN), Columbia University (2018). Global High Resolution Population Denominators Project - Funded by The Bill and Melinda Gates Foundation (OPP1134076).*

---

## Author Response (AR3)

**Spatial accessibility of emergency medical services under inclement weather: A case study in Beijing, China**

**Ref: nhess-2022-218**

We would like to thank the editor and reviewers for the thorough reading of the manuscript and the valuable remarks that helped us to improve the manuscript. We have revised the manuscript carefully according to the reviewer's comments, and have incorporated the suggestions into the revised manuscript.

The notes below provide a point-by-point response to each comment from the referees. The texts with blue font are the reviewer's original comments, the texts with black font are authors' responses. We have incorporated most of the suggestions made by the reviewers. Those changes are highlighted within the manuscript. If there is any question addressed unclearly or unsatisfied, we are always willing to make a revision based on reviewer's comments. Thank you again for the opportunity to be considered for publication in *Natural Hazards and Earth System Sciences*.

**Referee #1**

I thank the authors for addressing my comments. I think that the scoping for this paper, in the introduction and the conclusion, should be improved:

Introduction (Line 103-105): "Due to insufficient recorded traffic data, relatively few studies have been performed to analyze the impact of road access capacity on EMSs accessibility according to actual traffic speed variation.".

This claim is used to prepare the reader to the key components of this paper. However, it is not backed up with key citations. What are these few studies, their limitations, etc., so that the proposed one here is the "a first attempt to analyze the spatial accessibility of EMSs under inclement weather based on city-scale ground truth traffic data and meteorological data, where the former is usually difficult to obtain" [Lines 471-479]?

Conclusion (Lines 471 to 479): "To the best of the authors' knowledge, this study provides a first attempt to analyze the spatial accessibility of EMSs under inclement

weather based on city-scale ground truth traffic data and meteorological data, where the former is usually difficult to obtain. In previous literature, simulation methods were widely used on the research on EMSs accessibility or traffic capacity under inclement weather. The ground-truth traffic data that covers every road in the whole city, was hardly used in the previous studies of the impact of weather on traffic and accessibility. Our study could be a good empirical verification in this field of study. The reduction extent of EMSs accessibility was comparable to previous studies (Yin et al., 2021; Coles et al., 2017)."

This contribution should be crystal clear on the METHODOLOGICAL difference between the works of (Yin et al., 2021; Coles et al., 2017) and what is proposed here. While the case study sites/cities are different, the authors claim that this study is "… a first attempt to…" but then also claim that the results "…was comparable to previous studies (Yin et al., 2021; Coles et al., 2017)".
Overall, the specific METHODOLOGICAL added value should stand out for readers.

**Response**: We thank for reviewer's comment. We apologize for any misunderstandings that may have been caused by the comparison between our work and previous studies. We have provided additional explanations for this.
In the work of Yin et al., 2021, they simulated urban waterlogging scenarios under different rainfall intensities and the traffic speed was set based on recorded average traffic speed under normal conditions, the traffic speed variations induced by precipitation were not considered. While in our work, the traffic speeds were set based on the real-time traffic data under precipitation scenarios, the relationship between traffic speed and precipitation was further explored.
To the best of our knowledge, our work is the first attempt to analyze the spatial accessibility of EMSs under inclement weather based on city-scale ground truth traffic data and meteorological data. It is indeed inappropriate to compare our results with previous studies (Yin et al., 2021; Coles et al., 2017), since they did not consider traffic speed variations under precipitation. It is not to say that the results of our work should be exactly consistent with the results of similar studies by previous scholars, but in fact, they cannot be completely consistent because research areas and the methods are not the same.

We revised the description of Yin et al.'s work (2021) to better introduce their methods, please refer to Lines 94-99, page 5:
*Yin et al. (2021) assessed the vulnerability of EMSs to surface water flooding in*

*Shanghai, China by quantifying accessibility in terms of service area, population coverage and response time. They simulated urban waterlogging scenarios under different rainfall intensities and set traffic speed based on recorded average traffic speed under normal conditions, which didn't consider the traffic speed variations induced by precipitation.*

To avoid possible misunderstandings, we deleted the comparisons and elaborated the description of existing studies, please refer to Lines 484-494, page 24:

*To the best of our knowledge, there was no studies have been performed to analyze the impact of road access capacity on EMSs accessibility under inclement weather according to actual traffic speed variation. Our study provides an attempt to analyze the spatial accessibility of EMSs under inclement weather based on city-scale ground truth traffic data and meteorological data, where the former is usually difficult to obtain. In previous literatures (Yin et al., 2021; Coles et al., 2017; Albano et al., 2014), simulation methods were widely used on the research on EMSs accessibility or traffic capacity under inclement weather; however, the ground-truth traffic data that covers every road in the whole city under precipitation scenarios, was hardly used in the previous studies of the impact of weather on traffic and accessibility. Our study could be a good empirical verification in this field of study.*

**Referee #2**

The manuscript is investigating the impact of rainfall events on EMS, which is an interesting topic to study.

The authors have compared the traffic speed reduction during rainy days and calculated average speed reduction, which is used for further analysis. However, there are a number of assumptions or details not properly explained in the manuscripts.

1. The methodology uses the precipitation as the criteria for defining the wet days, the rainfall-runoff process is not directly reflected in the assumption. How are the influences of soil infiltration, natural and urban drainage in the process? Would 3 mm/2h exceed the design capacity of the sewer network in Beijing?

**Response**: We thank for reviewer's comment. We have checked the drainage design

standards in Beijing (DB11/ 685—2021, DB 11/T 1575—2018), which stipulate that the rainwater drainage design standard should not be less than a 2-year return period (about 55mm/2h). And a 3 mm/2h clearly would not exceed the design capacity of the sewer network in Beijing. However, even if precipitation of this intensity cannot cause road waterlogging, it can still lead to wet and slippery road surface and affect the drive speed. So, in our study, we used the actual recorded speed changes, which indeed showed that precipitation had an impact on traffic speed.

And we added some references to support this point:

*Chu L J, Fwa T F. Pavement skid resistance consideration in rain-related wet-weather speed limits determination[J]. Road materials and pavement design, 2018, 19(2): 334-352.*

*Katz, Bryan, et al. Guidelines for the use of variable speed limit systems in wet weather. No. FHWA-SA-12-022. United States. Federal Highway Administration. Office of Safety, 2012.*

To make it clear, we have revised the expression in Lines 159-166, pages 7-8:

*In this study, we set a rule that if the precipitation of more than 10 grids (over 5% area of the city) in Beijing is greater than 1.5 mm in 2 hours, it is considered a precipitation event. This amount of precipitation may not high enough to cause the rainfall-runoff exceed the drainage capacity of the sewer network in Beijing (DB11/ 685—2021, DB 11/T 1575—2018). But the precipitation would cause slippery roads and decrease in drivers' visibility, which would lead to a reduction of traffic efficiency and accessibility (Chu and Fwa, 2018; Katz et al., 2012).*

2. It is unclear if the calculations were done by only comparing the reductions during the 2h window or by the selected wet days. If it is done by every 2h window, how to reflect the delayed responses of runoff? If it is done by daily average, for the periods without rainfall during the day, were the traffic conditions the same to normal days? Should those periods be included?
   Should the temporal changes of rainfall affect the traffic condition? How is such a situation reflected in the methodology?

**Response**: We thank for reviewer's comment. Yes, we did the calculations by the 2h window of morning rush period. The precipitation is calculated as the cumulative precipitation within 2 hours of the morning rush period, and traffic speed is averaged by the speed during 2 hours of the morning rush period. Both are not daily averages. We apologize for the unclear definition of traffic speed in equation (1), we added more

explanation in Lines 220-233, pages 10-11. The original equation (1) was split into two equations, to make it clearer.

The original equation (1):

$$r_c = \frac{v_p - \frac{\sum_{j=0}^{m} v_{d_j}}{m}}{\frac{\sum_{j=0}^{m} v_{d_j}}{m}} \quad (1)$$

The new equations and their descriptions:

$$r_c = \frac{v_p - v_b}{v_b} \qquad (1)$$

*where $r_c$ is the traffic speed reduction rate in the selected period of the precipitation day to its corresponding baseline day; $v_p$ is the traffic speed in the selected period of the given precipitation day, and it is the average of the real-time traffic speed in every 2 minutes during the selected time period in that day; $v_b$ is the traffic speed in the selected period of the baseline precipitation days, which is calculated by eq.(2):*

$$v_b = \frac{\sum_{j=0}^{m} v_{d_j}}{m} \qquad (2)$$

*where $v_{d_j}$ is the traffic speed in the selected period of a baseline day, and it is the*

*average of the real-time traffic speed in every 2 minutes during the selected time period in that day; m is the number of baseline days. In this case, m equals 4. The average traffic speed reduction rate is obtained by averaging the reduction rates of all roads with reduced speed in the city.*

The reviewer's proposal of "the delayed effect of rainfall on runoff formation" is indeed worth considering and analyzing. In our study, more attention is paid to the immediate effects of rainfall, such as causing slippery roads and reduced visibility, without considering the cumulative effects of rainfall runoff. In further research in the future, maybe we can explore the difference in the impact of accumulated rainfall before the selected time window and within the time window on traffic speed.

Besides, we agree that the temporal variation of rainfall may affect traffic conditions. In this study, the problem was simplified and analyzed without considering the impact of changes within the 2-hour time window, as the selected 2-hour time window was not quite long, and the time resolution of the original precipitation data was only half an hour. If precipitation data with higher spatiotemporal resolution can be obtained in the future, the relationship between rainfall variation and traffic changes can be further

refined

In response to these points, we supplemented them in the discussion in Lines 502-504, page 25:

*First, we averaged the traffic speed reduction rate of all the roads in the city, as well as the precipitation data, which could conceal congestion hotspots. Besides, all the calculation was done by the 2-hour selected period, which may neglect the delayed responses of rainfall runoff and temporal variation of rainfall. In further studies, with higher resolution precipitation, along with corresponding traffic data, we could narrow the scale to blocks, pay more attention to local congestions, and analyze the correlation of precipitation and traffic speed on a finer scale.*

3. According to Figure 4, the reductions are very different from road to road for different rainfall conditions. Should the main roads and minor roads use the same average reduction in the analysis? Same question goes to roads in city center and rural areas. The single average value will underestimate the reductions in those areas that were impacted most.

**Response**: We thank for reviewer's comment. We agree that there are significant differences in the variation of traffic speed between roads. However, because the spatial resolution of precipitation data is relatively coarse, it is difficult to further refine the scale when analyzing the correlation between precipitation and traffic speed reduction. So, we average the speed reduction of all roads when comparing the overall speed reduction rate of the city.

But we did not do any average calculation of traffic when we build the road network and set speed for every road to analysis the accessibility. We did not put the average speed reduction into the analysis of accessibility, instead, the traffic speed of every road was set by the real traffic speed of the chosen date and chosen period.

To better illustrate this point, we have added the following explanation in Lines 241-243, page 11:

*In this study, the time needed to pass through the road is calculated by the length of each road divided by its corresponding traffic speed, and the service area analysis is carried out with time as the impedance. In different scenarios, the time impedance varies, since the traffic speed of each road is set according to the real-time traffic speed record of the chosen date and chosen period.*

4. In Figure 9, the delay of EMS is multiplied by the population, this is probably over exaggerating the information. It is unlikely all populations in the area require EMS

simultaneously. On the other hand, the capacity of EMS in different areas should also be considered. What if a hospital is exceeding its capacity, will the delay be further extended?

**Response**: We thank for reviewer's comment. We agree that not all populations in the area require EMS simultaneously. However, the busyness of emergency services in one region is roughly proportional to its population. We only use population as a weight coefficient, and the numerical value has no practical significance.

To better illustrate this point, we have added the following explanation in the methods section in Lines 280-283, page 13:

*The total transfer time is introduced to quantify the cumulative transfer time for each population grid based on its population size, which is the number of potential users of EMSs. The total transfer time is defined in this study by the shortest transfer time of each population grid to the nearest hospital multiplied by its population. The numerical value has no practical significance, and is only used for comparing the spatial differences among regions.*

Besides, we agree that the capacity of hospitals should be considered, unfortunately we are unable to obtain data like the number of beds, number of medical staffs, and medical material reserves of each hospital. If detailed public hospital data can be obtained in the future, this analysis can be further improved. Therefore, in our study, we assumed that the EMSs needs would not exceed the hospitals' carrying capacity. And we added this assumption in Lines 203-205, page 9:

*(6) The hospitals' carrying capacity is not been considered in this study, and we assume that the demand of EMSs would not exceed the first aid stations' and hospitals' carrying capacity.*